# Comparison of measurements from different radio-echo sounding systems and synchronization with the ice core at Dome C, Antarctica

**Anna Winter[1], Daniel Steinhage[1], Emily J. Arnold[2], Donald D. Blankenship[3], Marie G. P. Cavitte[3], Hugh F. J. Corr[4], John D. Paden[2], Stefano Urbini[5], Duncan A. Young[3], and Olaf Eisen[1,6]**

[1]Alfred-Wegener-Institut Helmholtz-Zentrum für Polar- und Meeresforschung, Bremerhaven, Germany
[2]Center for Remote Sensing of Ice Sheets, Lawrence, KS, USA
[3]University of Texas Institute for Geophysics, Austin, TX, USA
[4]British Antarctic Survey, Cambridge, UK
[5]Istituto Nazionale di Geofisica e Vulcanologia, Rome, Italy
[6]Fachbereich Geowissenschaften, Universität Bremen, Bremen, Germany

*Correspondence to:* Winter, Anna (awinter@awi.de)

**Abstract.** We present a compilation of radio-echo sounding (RES) measurements of five radar systems (AWI, BAS, CReSIS, INGV and UTIG) around the EPICA Dome C (EDC) drill site, East Antarctica. The aim of our study is to investigate the differences of the various systems in their resolution of internal reflection horizons (IRHs) and bed topography, penetration depth, and capacity of imaging the basal layer. We address the questions of the compatibility of existing radar data for common interpretation, and the suitability of the individual systems for reconnaissance surveys. We find that the most distinct IRHs and IRH patterns can be identified and transferred between most data sets. Considerable differences between the RES systems exist in range resolution and depiction of the bottom-most region. Considering both aspects, which we judge as crucial factors in the search for old ice, the CReSIS and the UTIG systems are the most suitable ones. In addition to the RES data set comparison we calculate a synthetic radar trace from EDC density and conductivity profiles. We identify ten common IRHs in the measured RES data and the synthetic trace. We then conduct a sensitivity study for which we remove certain peaks from the input conductivity profile. As a result the respective reflections disappear from the modeled radar trace. In this way, we establish a depth conversion of the measured travel-times of the IRHs. Furthermore, we use these sensitivity studies to investigate the cause of observed reflections. The identified IRHs are assigned ages from the EDC's time scale. Due to the isochronous character of these conductivity-caused IRHs, they are a means to extend the Dome C age structure by tracing the IRHs along the RES profiles.

## 1 Introduction

To predict the future evolution of ice sheets, the knowledge of their past response to climate changes is inevitable. Ice cores are valuable archives to study the climate of the past (Augustin et al., 2004). In contrast to other climate archives, they contain actual paleo-atmosphere in the form of air bubbles and hydrates that are trapped in the ice. With this advantage, they are the only means to answer some important questions in climate studies with respect to greenhouse gases, e.g., why did the glacial–interglacial cycles change from 40 ka to 100 ka at the mid-Pleistocene transition (MPT) and what drove the 40 ka cycles (Raymo et al., 2006)? For this reason the International Partnerships in Ice Core Sciences (IPICS) included the retrieval of the "Oldest Ice" ice core as one of their scientific goals (IPICS, 2006). This core should contain a continuous ice core record over at least 1.2 Ma, preferably 1.5 Ma, at some distance above the ice–bedrock interface. As compared to the oldest continuous ice currently on record (retrieved at Dome C (Augustin et al., 2004) and has an estimated age of 800 ka), this future core is likely to include the MPT and some 40 ka cycles.

There are only few promising regions for such an Oldest Ice ice core, all of them located close to dome or saddle positions on the East Antarctic Plateau (Fischer et al., 2013). As

many conditions have to be fulfilled at a site for old ice to exist and, equally important, to be retrievable in an analyzable way, extensive pre-site surveys are necessary to fill in gaps in the already existing data sets. Of great importance are not only ice thickness and internal structure, but also surface and basal mass balance, ice flow history, as well as temperature profile and geothermal heat flux. Since not all of these parameters are easy to determine in the field, modeling studies will be engaged to constrain upper and lower bounds on parameters which cannot be measured. Radio-echo sounding (RES) is a widely used method to investigate accumulation, ice thickness and internal structure (e.g., Urbini et al., 2008; Rodriguez-Morales et al., 2014; Mac-Gregor et al., 2015a). Variations in density, conductivity, or crystal orientation fabric (COF) cause the partial reflection of electromagnetic-wave energy, and thus appear as internal reflectors (internal reflection horizons, IRHs) in radargrams. The IRHs from changes in density and conductivity are formed at the same time near the surface (Vaughan et al., 1999; Dowdeswell and Evans, 2004) and then advected by compaction and ice flow. Density variations are the primary cause for IRHs in the uppermost few 100 m of the ice sheet, but do not occur in deeper parts (Millar, 1981). The IRHs from conductivity changes, in contrast, can be found throughout the ice sheet. The reflection coefficients of those IRHs are related to changes in the imaginary part of the complex dielectric permittivity and are thus proportional to conductivity changes and inversely proportional to frequency. A change of crystal orientation fabric (COF) is the second reason for reflections in the deeper parts of the ice column, predominating in zones of high shear. In RES measurements the conductivity-based IRHs can be distinguished by the frequency dependence of their reflectivity from IRHs caused by COF and density, which have frequency independent reflection coefficients related to changes in the real part of the complex permittivity (Fujita et al., 1999, 2000). Conductivity itself was assumed frequency independent in the range of RES frequencies, but more recent work implies that its frequency dependence cannot be neglected for e.g., attenuation studies (MacGregor et al., 2015b).

RES data provide information about the englacial age structure away from any ice core. The isochronous conductivity-based IRHs can be traced continuously over long distances in the ice sheets (e.g., Steinhage et al., 2013), and thus be used for extrapolating the age-depth distribution from ice cores along the RES profiles. In comparison with ice core data sets this larger amount of information about the age structure of ice sheets is useful for the evaluation of ice-flow models and an important criterion at the current stage of the models (Sime et al., 2014). Furthermore, IRHs are used to directly derive ice dynamics (e.g., Karlsson et al., 2012; Winter et al., 2015) and attenuation rates, whereof depth-averaged temperatures can be obtained (Matsuoka et al., 2010; MacGregor et al., 2015b; Jordan et al., 2016).

The great value of RES data for the investigation of the ice sheets has led to numerous campaigns with gradually more sophisticated radar systems over the decades. Due to the variety of differently motivated surveys and the rapid technical development, the data of one particular radar system usually is confined to a more or less small area. Therefore it is inevitable to combine data of different systems when working on larger-scale or even ice-sheet-wide problems. E.g., Cavitte et al. (2016) use data of two different radar systems to obtain a continuous connection of the East Antarctic deep drill sites Dome C and Vostok, and a third system for an alternative connecting route and separate quality control. Searching for the Oldest Ice, it would be beneficiary to include the existing measurements of even more radar systems. However, it neither is clear nor has it been investigated if the data, measured with different radar systems at different times and having different characteristics, are comparable and can be assembled to one data set suitable for this purpose or if there are less confidable data sets.

In this study we address these questions and, for the first time, compile the data of five different RES systems, all recorded in close vicinity to the EPICA Dome C (EDC) drill site, for comparison of the various systems' strengths and weaknesses. Furthermore, we calculate a synthetic radar trace, using the Dome C ice core profiles of density and conductivity, as established by Eisen et al. (2004) and Eisen et al. (2006) to relate the radar measurements to ice properties. The modeled radar trace is also used for an accurate depth inversion and thus reliable age assignment of 10 horizons, identified in the RES data. As the EDC core yielded the oldest continuously dated ice so far, it is a good starting point for tracing the oldest possible layers.

## 2 Synthetic radar traces

To relate measured RES data to the physical properties and age of the ice, synthetic radar traces are calculated from measured ice core data. The modeled traces are then compared to the measured RES data with respect to the questions of depth origin and nature of the RES reflections. In the sections below we describe the ice core data used for calculating the synthetic traces (Sect. 2.1) and the actual modeling (Sect. 2.2). In Sect. 2.3 we describe how we derive the value for the permittivity of ice that we use in all further proceedings.

### 2.1 Ice core data

We use the records of the second EDC ice core (EDC99) that was drilled in the austral seasons 2000–2004. The drill site is located on the East Antarctic Plateau at $123.35°$ E and $75.10°$ S, 3233 m above sea level. It has a yearly accumulation rate of $25 \text{ kg m}^{-2} \text{ a}^{-1}$ and a mean annual surface temperature of $-54.5°$C. The ice thickness at this location is $3309 \pm 22$ m (The EPICA Dome C 2001-02, science and drilling teams, 2002). In 1999, the first drilling attempt had to be abandoned

because the drill got stuck in a depth of almost 800 m. In the second attempt a depth of 3260 m was reached, with only a few meters missing to the bedrock (Augustin et al., 2004). The core was dated back to roughly 800 ka BP by e.g., Bazin et al. (2013), and thus comprises the oldest continuously retrieved ice to date.

Dielectric profiling (DEP, Moore (1993)) measurements on the core were conducted in the field at temperatures of $-20 \pm 2°C$ and a frequency of 100 kHz. The data set consists of conductivity values ($\sigma$ in $\mathrm{S\,m^{-1}}$) for the depth range of 6.8 m to 3165.2 m with a resolution of 0.02 m. The data were corrected to a temperature of $-15°C$ and cleaned of data points where the core was broken (Parrenin et al., 2012; NOAA, 2011). The record is extended up to the surface by linear interpolation to a value of 4.05 $\mu\mathrm{S\,m^{-1}}$. Gaps due to removed data points are also linearly interpolated and the record is linearly resampled to 5 mm.

The density ($\rho$ in $\mathrm{kg\,m^{-3}}$) of the EDC99 core was measured with the $\gamma$-absorption method, which is also described in Eisen et al. (2006), at the Alfred Wegener Institute, Bremerhaven, Germany for the depth range of 6.8 m to 112.7 m in 1 mm increments (Hörhold et al., 2011). Gaps in the record are linearly interpolated and the record is also resampled to 5 mm. For depths outside the measurement range the density is logarithmically extrapolated up to the density of ice, $\rho_{ice} = 917\ \mathrm{kg\,m^{-3}}$.

The records for density and conductivity are then combined to one record of depth, density and conductivity from the surface to 3165.2 m depth in 5 mm increments.

## 2.2 Electromagnetic modeling of radar traces

Radar measurements are recorded in the two-way travel time (TWT) domain. The reflection peak of a reflector of a certain depth is recorded after the time a transmitted wave needs to travel to the reflector and back again. To convert the depth profile of our combined ice core record to the TWT domain of the RES data, we need the depth-dependent electromagnetic wave speed in firn and ice

$$c(z) = \frac{c_0}{\sqrt{\varepsilon'(z)}}, \tag{1}$$

with the vacuum wave speed $c_0$ and the real part $\varepsilon'$ of the complex relative dielectric permittivity

$$\varepsilon^* = \varepsilon' - i\varepsilon'' = \varepsilon' - i\frac{\sigma}{\varepsilon_0\omega}, \tag{2}$$

where $\sigma$ is the conductivity, $\varepsilon_0$ the vacuum permittivity and $\omega$ the circular frequency. For readability we use "permittivity" hereafter for the real part of the complex permittivity ($\varepsilon'$).

As the accuracy of the DEP measurement performed at EDC does not allow for an inversion of the complex-valued permittivity of a two-phase mixture, as described by Eisen et al. (2006), we use the real-valued dielectric mixture equation by

Looyenga (1965) to calculate $\varepsilon'(z)$ from $\rho(z)$:

$$\varepsilon'(z) = \left( \frac{\rho(z)}{\rho_{ice}} \left( \varepsilon'^{\frac{1}{3}}_{ice} - 1 \right) + 1 \right)^3, \tag{3}$$

with the measured density $\rho(z)$ and the pure-ice values for density and permittivity $\rho_{ice} = 917\ \mathrm{kg\,m^{-3}}$ and $\varepsilon'_{ice} = 3.17$. In Sect. 2.3 we describe how we derived the value for the permittivity of ice for our study. Below the depth, where the density of ice is reached we use the constant permittivity $\varepsilon' = \varepsilon'_{ice}$. It should be noted at this point that only conductivity-caused reflections and no permittivity-caused (i.e. COF and density based) reflections can be modeled in this way (Fujita and Mae, 1994). Neglecting the complex character of the relative permittivity in the two-phase mixture leads to errors in the complex permittivity, especially in the firn (Wilhelms, 2005). However, for the purpose of reproducing the signature of the conductivity-caused IRHs as measured by radar, not the absolute value but the changes of conductivity are important. Though reflections occur at the wrong TWTs in the synthetic trace when incorrect real permittivities are used in the model, we avoid these errors by calibrating the synthetic with the measured radar trace (see Sect. 2.3).

The permittivity record is smoothed with a 0.2 m running mean to prevent the masking of the conductivity-induced reflections and the too quick reduction of the propagating energy in the synthetic radar trace by a multitude of permittivity-induced reflections in the firn section. More extensive reasoning and effects of this procedure can be found in Eisen et al. (2006).

Permittivity and conductivity, processed as described above, are input parameters for the 1D-FD (One-Dimensional Finite Difference) version of the model "emice" (Eisen et al., 2004) that calculates synthetic radar traces by solving Maxwell's equations. The depth increment of the model domain is 0.02 m. The maximum depth is 3165.2 m and an absorbing boundary is implemented in the depth direction. The time increment is 0.02 ns, which fulfills the Courant Criterion that ensures the stability of the numerical calculations (Courant et al., 1928; Taflove and Hagness, 1995).

Following Eisen et al. (2006), we use a source wavelet of two and a half 150 MHz cycles. It should be noted here, that, for simplicity this wavelet is based on the burst and pulse radar systems rather than the chirp systems, which require additional post processing like pulse compression (Sect. 3). However, this synthetic pulse is much shorter and the wavelet is not identical to any of the RES system ones. We chose it as trade-off between being long enough to reproduce some interference effects, but relatively short for determining reflector depth in sufficient resolution.

The envelope of the calculated trace corresponding to the reflected energy is obtained by conducting a Hilbert magnitude transformation (Hilbert, 1906; Taflove and Hagness, 1995). Finally, the trace is smoothed with a Gaussian running mean

of 100 ns. The result of this step is the synthetic trace that we use for comparison with the measured RES data, as described in Sect. 5.

### 2.3 Assessing the permittivity of ice

To calculate the correct TWTs for reflectors in our synthetic radar trace we have to use the correct permittivities. For too small permittivities the wave speed is too high and a distinct reflection does thus appear too early (Eq. 1). This time shift increases with the absolute depth of the reflector. As the real permittivity could not directly be measured at Dome C, we are looking for an average value for the permittivity below the firn–ice transition $\varepsilon'_{ice}$ that best reproduces the reflection TWTs compared to measured RES data. Above the firn–ice transition we use measured densities to calculate permittivities, as described in Sect. 2.2. As reference RES data we choose the AWI data for their small distance to drill site, high vertical resolution and being a burst system which is closer represented by the source wavelet of the model than the chirp systems (see Sect. 3 on RES data). Note that it is not our aim to get the exact value of $\varepsilon'_{ice}$ but rather a good estimate with this method, so we can easily match reflection peaks of all measured RES data with the synthetic trace in a later step. The exact permittivity is not needed throughout our study because we do not use a velocity function to calculate the depths of RES IRHs but a sensitivity study with the synthetic trace (Sect. 5.3).

With the trial-and-error method we compare the synthetic traces of model runs with different $\varepsilon'_{ice}$ to the AWI trace, starting with the commonly used value of $\varepsilon'_{ice} = 3.15$ (e.g., Rodriguez-Morales et al., 2014). We compare the TWTs of ten distinct reflections distributed between approximately 2.3 $\mu$s and 24.3 $\mu$s TWT between synthetic and AWI trace. This synthetic trace shows smaller TWTs than the measured one, with increasing time lags towards greater TWTs. For this reason we repeat the procedure with an $\varepsilon'_{ice}$ increased by 0.01, and so on. The best result is obtained with $\varepsilon'_{ice} = 3.17$ for which we do not get TWT lags that are systematically changing with increasing TWT between synthetic and measured radar traces and for the compared IRHs. Therefore we conclude that 3.17 is a suitable estimate for $\varepsilon'_{ice}$ in our study region and we will use the synthetic trace calculated with this value for our further proceedings. This value is also found reasonable by Bohleber et al. (2012) for slightly anisotropic configurations and is close to the pure isotropic ice value of $\varepsilon'_{ice} = 3.16$ found in their laboratory experiments.

### 3 Radio-echo sounding data

The profiles closest to the drill site were selected from the RES data available in the Dome C area. The distance between the drill site and the furthest profile is less than 2 km. The positions of the RES profiles relative to Dome C are shown

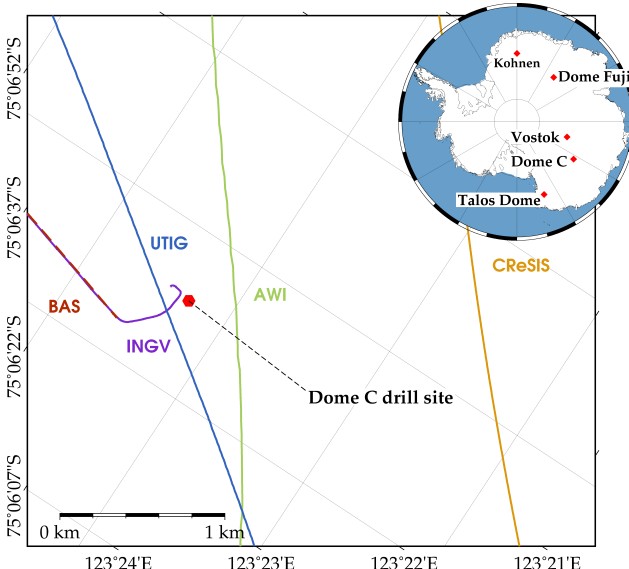

**Figure 1.** The deep drill sites in East Antarctica and close-up to Dome C with the RES profiles. The crosses mark each profile's closest trace to the drill site. The location of the EDC drill site is marked by the red hexagon.

in Fig. 1. As the influence of different radar systems on the recorded radargram is to be examined, we use the data of five different institutes: the Alfred Wegener Institute (AWI), Bremerhaven, Germany, the British Antarctic Survey (BAS), Cambridge, UK, the Center for Remote Sensing of Ice Sheets (CReSIS) at the University of Kansas, Lawrence, USA, the Istituto Nazionale di Geofisica e Vulcanologia, Rome, Italy (INGV) and the University of Texas Institute for Geophysics (UTIG), Austin, USA. The characteristics of the different systems and the data processing are described next. The system characteristics are summarized in Table 1.

### 3.1 AWI

The airborne radar system of AWI is a burst system with a carrier frequency of 150 MHz that was operated in toggle mode with 60 ns and 600 ns bursts (Nixdorf et al., 1999). Measurements were conducted in austral season 2007/08 with the DC-3T aircraft Polar 5. We use the data with 60 ns bursts and stack it 10-fold. The stacked data have a trace distance of about 75 m and a vertical sampling interval of 13.33 ns. The profile passes the Dome C drill site at a distance of 280 m.

### 3.2 BAS

The BAS profile was recorded in season 2005/06 with an airborne radar system on a Twin Otter in 450 m distance to the drill site. The source is a 4 $\mu$s chirp wavelet with a center frequency of 150 MHz and a bandwidth of 10 MHz. The vertical sampling interval is 45.45 ns. The data are chirp

compressed and a horizontal smoothing with a 49 sample moving-average filter and 10-fold stacking is applied. The trace distance after stacking is 45 m.

### 3.3 CReSIS

CReSIS had one campaign in the Dome C area in season 2013/14, using the Multi-Channel Coherent Radar Depth Sounder (MCoRDS) on an Orion P3 aircraft (Rodriguez-Morales et al., 2014). The source wavelets are 1, 3, and 10 $\mu$s chirps, each running linearly through the frequency range of 180 MHz to 210 MHz (Gogineni, 2012). We use the L1B-data CSARP_standard file, processed with pulse compression, focused SAR processing, and array processing with multilooking (CReSIS, 2016). The final product has a vertical sampling interval of 33 ns and a trace distance of 30 m. The profile passes the drill site at 1745 m distance.

### 3.4 INGV

The INGV profile was measured in December 2011 during a test of a 200 ns pulse radar system with a carrier frequency of 150 MHz, recording the envelope only. The horizontal trace distance is about 0.25 m and the vertical sampling interval is 40 ns. The 2.7 km long profile passes the drill site in 65 m distance. A spiking deconvolution, low pass filtering and gain adjustment is conducted. We stack the data 10-fold.

### 3.5 UTIG

The radar profile of the UTIG was collected with the High-Capability Radar Sounder (HICARS) in season 2008/09 from a DC-3T aircraft in 150 m distance to the drill site. The system uses a 1 $\mu$s chirp wavelet running linearly through the frequencies from 52.5 MHz to 67.5 MHz. The HICARS system is described in Peters et al. (2005) and the SAR processing in Peters et al. (2007). The recorded data were filtered with a 10 MHz band notch filter and a convolution. An automatic gain control is conducted and the data are stacked horizontally coherently ten times, log detected and incoherently stacked five times so the final trace rate is 4 Hz. This gives a trace distance of about 22 m for this product. The vertical sampling interval is 20 ns (Young et al., 2011; Cavitte et al., 2016).

### 4 Assembling the data sets

Different system characteristics and processing result in different appearance of the RES data. Our aim is to compare the RES and synthetic radar data in terms of identifying distinct reflectors that can be found in all data sets and that can confidently be matched in between the different data sets. Our basis for determining the origin of the IRHs in the RES data is by relating them to the conductivity record. For this reason, we neglect the first few microseconds, i.e., the upper

few hundred meters, where most reflections are due to density changes. Like Karlsson et al. (2016), we found that this matching can best be achieved manually. But additionally we use a combination of two different ways of imaging:

1. Single traces as reflected energy versus TWT (A-scope) to compare the reflection peaks' shapes and positions. For every RES profile, the trace (of stacked data) closest to the EDC drill site is selected and plotted as a single trace. This is described in Sect. 5.1 and shown in Fig. 2. To make the different data comparable, we first shift them in time so that the surface reflections are at TWT zero. Here, we use the maximum of the surface reflection peaks for the systems with chirp wavelet (BAS, CReSIS and UTIG), but its steepest slope for the pulse systems. This is motivated by the systems differing in signal generation and digitization. The depth of a reflector is depicted by the maximum of its reflection for the chirp systems, but by the rise of its reflection for pulse systems. Note that this is also valid for internal reflections. The position of the surface-reflection pick for all traces is marked by the vertical black line in the left panel of Fig. 2. The exponential trend is removed from every trace. The peak amplitudes of the reflections decline in a different manner for the different data, depending on the source wavelet of the radar system and the processing. For that reason, we scale the data differently for the different depth sections to make potential reflections in the basal region more visible.

2. Radar profiles of several kilometer length around the closest trace to the drill site, plotted as TWT vs. trace number with amplitude values in gray scale (Z-scope), as shown in Fig. 3 and described in Sect. 5.2. This way of imaging is especially suitable to compare specific sequences of reflections and to check whether the reflections matched in the A-scope image are spatially coherent and representative over larger regions, e.g. for extrapolation. Again, the surface reflections are shifted to TWT zero and we plot the logarithm of the amplitudes for all RES data. For the deepest third (bottom panel) we use differently processed, i.e. 2D-focused, CReSIS and UTIG data for an improved visibility of the deep internal structure.

### 5 Results

In this section we compare the different RES data and the synthetic radar data in order to match some reflections or reflection patterns distributed over the depth range. In Sect. 5.3 we determine the depth origin of the identified horizons by means of sensitivity studies with the conductivity record.

### 5.1 Single traces, A-scope

Figure 2 shows the traces closest to EDC of all RES profiles, and the synthetic trace. The positions of the measured traces are marked by crosses in Fig. 1. The left panel of Fig. 2 shows the reflections of the air–ice interface, which marks

**Table 1.** Characteristics of the five RES systems. The fourth column gives either the bandwidth B in case of the chirp systems, or the pulse length in case of the pulse and burst systems. The sixth column gives the vertical resolution due to bandwidth (for distinguishing two reflectors), determined as $\frac{kc_0}{2B\sqrt{3.17}}$ for the chirp systems. For the window widening factor k we use 1.53, as given in CReSIS (2016). Further details on data recording and processing can be found in the referenced papers, where the data are published.

| System | aircraft | center frequency MHz | bandwidth/ pulse length | vertical sampl. freq. MHz | vertical resolution m | horizontal sampl. distance m | reference |
|---|---|---|---|---|---|---|---|
| AWI | DC-3T | 150 | 60 ns | 75 | 5.1 | 75 | – |
| BAS | Twin Otter | 150 | 10 MHz | 22 | 12.9 | 45 | – |
| CReSIS | Orion P3 | 195 | 30 MHz | 30 | 4.3 | 30 | CReSIS (2016) |
| INGV | ground based | 150 | 200 ns | 25 | 16.6 | 2 | Urbini et al. (2015) |
| UTIG | DC-3T | 60 | 15 MHz | 50 | 8.6 | 22 | Cavitte et al. (2016) |
| Modeled | – | 150 | 0.2 ns | 1000 | 8.4* | – | Eisen et al. (2004) |

*the vertical resolution of the synthetic data is controlled by the filtering of the trace as described in Sect. 2.2.

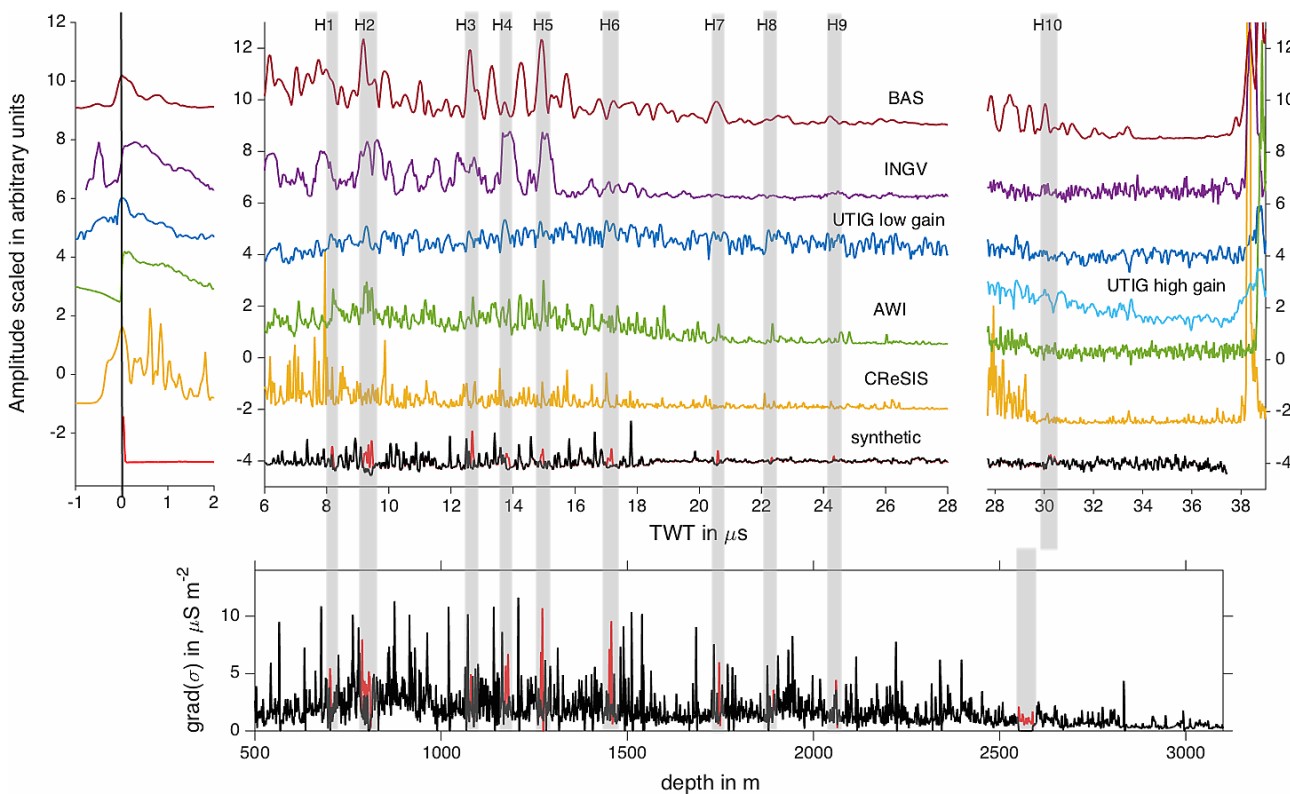

**Figure 2.** A-scopes for traces of the five radar systems and synthetic trace. The amplitudes are scaled individually and the exponential trend is removed. The surface reflection of each trace is shifted to time zero (left panel). Some distinct reflections that can be seen in some or all of the traces are gray-shaded. The bottom panel shows the envelope of the gradient of the conductivity profile. The peaks that are plotted in red cause the identified reflections H1–H10 in the synthetic trace.

TWT zero for each trace.

The upper middle part of Fig. 2 shows the traces for the majority of the ice column. In this section we find a number of distinct reflections that can be identified in some or all of the traces. We highlight ten of them (H1–H10), shaded in gray, for which we are confident to have them matched correctly and use them for further discussion. Those events are also used for the sensitivity studies with the conductivity record. It should be noted here that the peaks do not have the same relative amplitudes or width in the various data. Furthermore, not all IRHs can be found in all the data. The envelope of the gradient of the conductivity profile that is used for calculating the synthetic trace is plotted in the bottom panel of Fig. 2. The peaks of the gradient of the conductivity repre-

sent the greatest discontinuities in the dielectric properties, and thus are associated with the reflection peaks in the radar data. The parts of the profile plotted in red color are the conductivity signals that have to be removed for the identified reflections to disappear from the synthetic trace. In that way, we are able to assign the reflections with their depth and age, as described in Sect. 5.3. What is striking, when comparing the different RES traces, is the comparatively low vertical resolution of the INGV and BAS data. Multitudes of peaks from the other data are not separately resolved by the INGV and BAS systems, but combined to wider peaks. However, it is not always obvious, which peaks join together into one peak. Nevertheless, there are still reflections that are visibly similar to those in the other traces. E.g. the INGV and BAS peaks at about 15 $\mu$s (H5) can be matched with the one in the AWI data or the INGV peak just before 14 $\mu$s (H4) and the double peak at 17 $\mu$s (H6) with the UTIG data. In between these reflections, though, the appearance differs considerably from the other data.

There are small time shifts for the identified reflections in the different RES data. In the CReSIS trace, for example, the peaks are usually about 50–100 ns earlier than in the other traces. The time differences for the peaks in between the other data are much smaller.

The third panel shows the bed reflections of the measured RES data and, just before the bed reflection, the section of the basal layer. In this last section, a reflection can be found at about 30 $\mu$s that fits well in the CReSIS, UTIG high gain, INGV, BAS and synthetic data (H10).

The TWTs for the bed reflections fit well for AWI and UTIG data, with the UTIG reflections having a longer slope than the AWI, and the UTIG high gain a longer one than the low gain bed reflection. In the BAS, INGV and CReSIS trace the bed reflections occur a few hundred nanoseconds earlier. Possible reasons for the differences in the timing of the bed reflections and internal reflections are discussed in Sect. 6.1.2.

### 5.2 Radargrams, Z-scope

Figure 3 shows Z-scopes of the five RES profiles. Unlike in Fig. 2, here the TWT serves as the vertical axis. In the leftmost panel the synthetic trace is shown. In the second panel this trace is adjacently plotted 200 times with the amplitude in gray scale. White noise is superimposed on each synthetic trace to get the appearance of a measured radargram. The other panels show the measured radar data of the five radar systems, processed as described in Sect. 3. We use an approximately 5 km profile length for all of the images with the exception of the INGV profile, which is only 2.7 km long. Like with the single traces, again some reflectors can be matched nicely. Here, especially sequences of IRHs are striking. For example the three closely spaced reflectors at about 6.5 $\mu$s TWT in the synthetic radargram (red arrow in Fig. 3) that can also be found in the measured data, although with a slightly different appearance. In the AWI data, the first re-

flector is the most pronounced one, and in the UTIG data they are rather blurred into one broad reflector. Another nice example is the strong reflector just below 8 $\mu$s, followed by the wider sequence below 9 $\mu$s (H1 and H2, yellow double arrow) that catch one's eye in all of the RES data. Those can also be matched with events in the synthetic data, whereas in the latter, there exist more strong reflectors in between. Hereby, the one at 9 $\mu$s is especially distinct and has a counterpart only in the CReSIS data. The section from about 13 to 16 $\mu$s TWT (blue lines), with densely spaced, relatively strong reflectors, is also similar in all of the RES data. It starts with a double-reflection, corresponding to H3 in Fig. 2, that can also be found in the synthetic radargram. In the middle section of Fig. 3 the reflectors at 19.8 $\mu$s and 20.5 $\mu$s (H7) are the most striking ones in the synthetic data (light blue double arrow). The most alike counterpart of this sequence is to be found in the BAS data. But there is also a match in the other RES data. Notable is that the first of the two reflectors is more pronounced in the UTIG and INGV data, whereas it is the second in the CReSIS data.

Striking differences between the RES systems exist in the quality of recording reflections from the basal region, shown in the lowermost panels of Fig. 3. In the AWI data, where the IRHs are nicely resolved in the upper two thirds, the visibility of reflectors ceases at 28 to 29 $\mu$s. The only distinct reflector after that is the bed reflection 10 $\mu$s later, leaving about 800 m without IRHs. The same is the case for the INGV data. In contrast, IRHs are clearly visible down to about 33.5 to 34.0 $\mu$s TWT in the BAS, CReSIS and UTIG data, ending with a relative strong continuous reflector with strong vertical variation (green arrows). In the BAS data there even are some signals, spatially coherent for a few kilometers, as deep as approximately 36 $\mu$s TWT. The same can be found in the CReSIS and UTIG profiles a few tens of kilometers away from EDC, which are not shown here.

At the very bottom of the figure, the bed reflections of the RES data can be seen. These measured bed-reflection depths are not to be compared with the strong reflection at the bottom of the synthetic radargram. The latter marks solely the margin of the model, which is equal to the end of the DEP record. At least 100 m are missing from the synthetic trace to the actual bed, which makes a difference of more than 1 $\mu$s in TWT.

### 5.3 Depths of the RES reflectors

To determine the depths of the IRHs, identified in Fig. 2 and Fig. 3, we conduct a sensitivity study of the synthetic trace as described in Eisen et al. (2006). By sensitivity study we mean that we remove certain peaks from the measured conductivity profile (the gradient of which is shown in the bottom panel of Fig. 2) and run the model with the changed input conductivity profile. As a result the respective reflection peaks disappear from the synthetic trace. As the synthetic trace closely resembles the conductivity profile, the conductivity peaks of

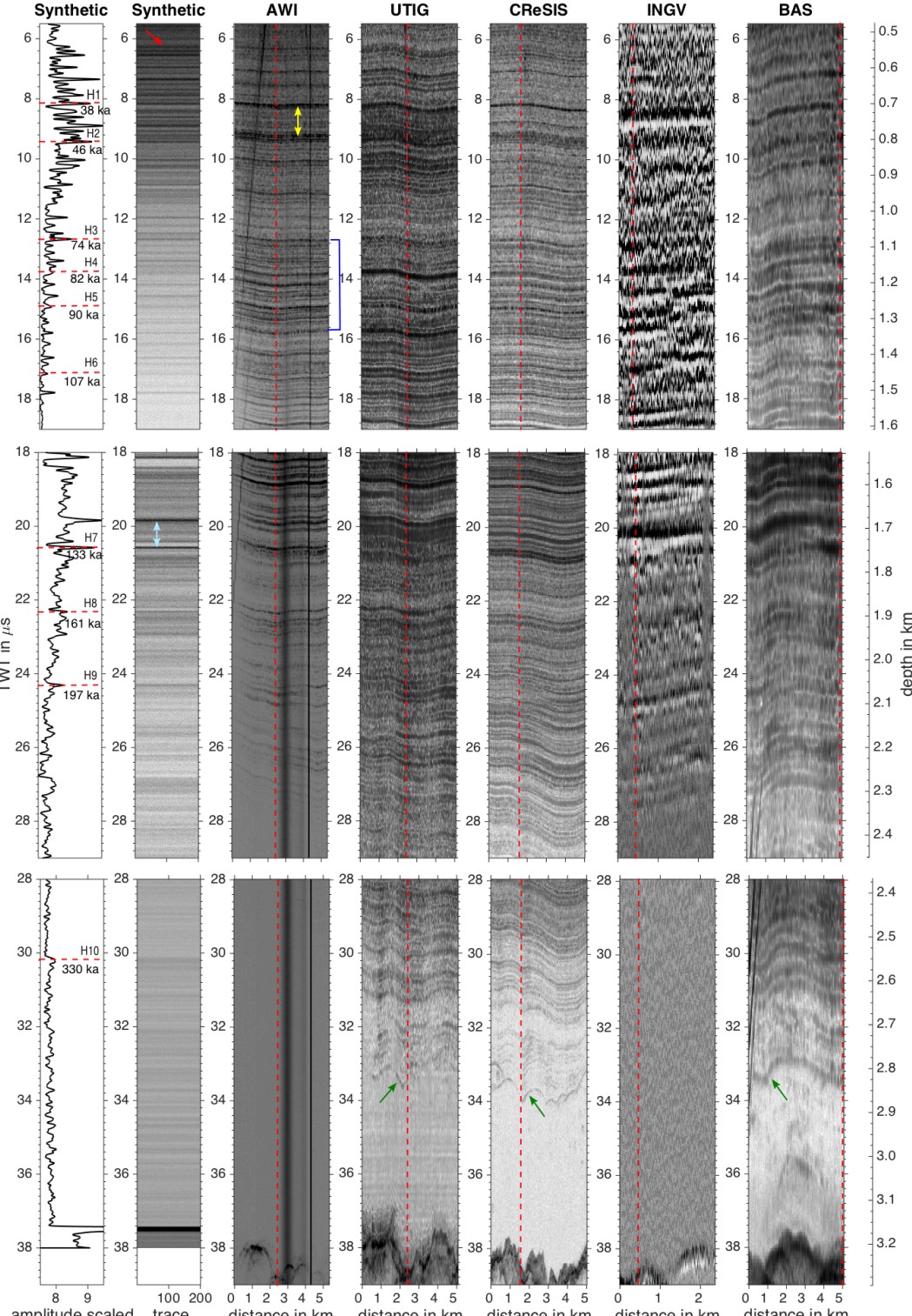

**Figure 3.** Z-scopes for synthetic and RES data sets of the five radar systems. The surface reflections are shifted to time zero as shown in Fig. 2 and the vertical red lines mark the positions of the traces of Fig. 2. The length of the RES profiles is indicated on the horizontal axes. For the depth axis we convert TWT to depth with a wave velocity of $c = 168.5$ m $\mu s^{-1}$ and a firn correction of 10 m. For the bottom UTIG and CReSIS panels a 2D-focused processing is applied. The colored arrows and lines mark distinct reflector patterns, closely described in Sect. 5.

interest can be identified with relatively small effort. An exception is the very uppermost part (~400 m), where the reflectivity is influenced not only by conductivity but also by density variations. The bottom trace of Fig. 2 shows the synthetic trace calculated with original (red) and changed (black) conductivity profile, and the bottom panel shows the gradients of the corresponding conductivity profiles. If a reflection in the RES data can confidently be matched with one of the synthetic radargram, the depth of the reflection can immediately be transferred from the conductivity profile. It has to be taken into account, however, that the determined depth is the horizon's depth at the ice core location, and may differ from its depth at the RES trace. Even so, the horizon can be assigned with an age. Due to the conductivity-induced IRHs being isochronous, their age is the same at the position of the RES profile, even if the depth is somewhat different. If matched correctly, the uncertainty of the reflector's age depends only on the width and number of the reflection-causing conductivity peaks and the accuracy of the age scale. The advantage of the sensitivity approach over converting TWT to depth using ice core densities and a velocity function is that the depth uncertainties do not accumulate with depth, but are independent of the absolute depth.

We remove sections from the conductivity record so that the gray-shaded reflections from Fig. 2 disappear from the synthetic trace. In that way we find that e.g., reflection H1 is caused by the conductivity peaks in the depth range of 700.54 m–702.64 m, or that the sequence H2 is caused by a multitude of conductivity peaks, spanning about 22 m depth. All identified reflections, the depth ranges of their inducing conductivity sections, and their age ranges with standard deviation according to the AICC2012 timescale (Veres et al., 2013; Bazin et al., 2013) are listed in Table 2. Age uncertainties for the IRHs due to reflector width or shifts in TWT are discussed in Sect. 6.2.2.

# 6 Discussion

In the sections below we discuss the results of our comparison of the different data sets. Section 6.1 addresses the comparison of the measured RES data among each other and gives possible reasons for differences between the data. In Sect. 6.2 we look at the connection of the RES data with the synthetic radar trace, and thus the ice core data, including an uncertainty assessment

## 6.1 Comparability of RES data

As pointed out in Sect. 5.1 and 5.2, there are some common features notable in all of the RES data and the synthetic trace (e.g., the strong IRH at 8 $\mu$s). Especially the AWI, BAS, CReSIS and UTIG radargrams show similar patterns of reflectors, like many densely spaced reflectors (e.g., starting from 13 $\mu$s), or of lacking reflectors (22 $\mu$s, 32 $\mu$s) at the same

depths. The distinct reflectors (that are usually chosen for tracing) can be identified and traced (at least for the lengths of the investigated profile sections) in those RES data. However, there are conspicuous differences, when comparing the data with respect to resolution and penetration depth. The differences and reasons are discussed in the sections below.

### 6.1.1 Vertical and horizontal resolution

There are dissimilarities in the RES data in markedness and vertical expansion of the reflectors. Those can partly be explained by the different range resolutions of the various radar systems, due to different source wavelets, receiver bandwidths, sampling rates and post-processing. The sampling intervals vertically range from 13.33 ns (AWI) to 45.45 ns (BAS) (see Sect. 3). This gives one sample every 1.1 m, and 3.8 m, respectively in the ice of the Dome C region. The vertical resolution due to source wavelet and received-antenna system bandwidth ranges from about 3 m to 17 m, as listed in Table 1. To some extent, the differences in the RES data can therefore be attributed to the varying range resolution of the different systems. The systems with lower range resolution are not able to capture multiple closely-spaced conductivity changes, and these closely-spaced layer variations are only represented by a single reflector (of potentially complex shape). Regarding the vertical resolution of IRHs at intermediate depths, we attest the AWI, CReSIS and UTIG systems to be the best quality, which is expected per the higher range resolution identified in Table 1. The CReSIS system shows the most detailed structure, while the AWI system has the least penetration depth of the three systems. Due to their lower vertical resolution, the INGV and BAS radargrams, but also the AWI and UTIG data, show examples (at 10 $\mu$s and 20 $\mu$s) where a series of reflectors in CReSIS data are depicted as only one wider reflection. The detailed structure in the CReSIS data is advantageous for synchronizing IRHs at a specific location in high resolution, like we do in this study. However, Cavitte et al. (2016), who use several radar data sets to connect the EDC and Vostok drill sites, point out that the high vertical resolution might make it difficult to trace the IRHs over wide distances because the IRHs can thin out more easily than for systems with lower vertical resolution and therefore more robust IRH returns. As for the compatibility of the data sets, we assume that there are no major issues in combining AWI, CReSIS and UTIG data at one location where the profiles are close to each other, due to their very similar reflector patterns. Yet, the additional vertical resolution of the CReSIS data could add ambiguity to combined data interpretations that include tracing of IRHs (Cavitte et al., 2016). A similar problem arises if the lower resolved BAS and INGV data are included, as it might become difficult to decide which reflector to continue with when going from lower to higher resolution data. In Fig. 4 we show the intersection of AWI and UTIG radargrams, the location of which can be inferred from Fig. 1. Two example IRHs (H1

**Table 2.** Identified layers of Fig. 2 and 3, their approximate TWTs, depth ranges of their inducing conductivity sections and corresponding age with age uncertainties (average of published AICC2012 age uncertainties of the top and bottom depth) on the AICC2012 timescale (Veres et al., 2013; Bazin et al., 2013).

| Horizon | TWT | depth top | depth bottom | age top | age bottom | uncertainty |
|---------|-----|-----------|--------------|---------|------------|-------------|
| | $\mu$s | m | m | ka | ka | ka |
| H1 | 8.0 | 700.54 | 702.64 | 38.17 | 38.30 | 0.58 |
| H2 | 9.5 | 786.84 | 808.80 | 45.49 | 47.22 | 0.78 |
| H3 | 12.5 | 1078.90 | 1081.36 | 73.66 | 73.96 | 2.00 |
| H4 | 13.5 | 1172.04 | 1179.06 | 82.03 | 82.58 | 1.53 |
| H5 | 15.0 | 1267.34 | 1271.30 | 90.04 | 90.40 | 1.60 |
| H6 | 17.5 | 1447.58 | 1458.16 | 106.32 | 107.49 | 1.88 |
| H7 | 20.5 | 1745.80 | 1746.02 | 132.74 | 132.77 | 2.13 |
| H8 | 22.5 | 1891.54 | 1892.98 | 160.96 | 161.24 | 3.50 |
| H9 | 24.5 | 2060.14 | 2060.40 | 197.17 | 197.23 | 1.96 |
| H10 | 30.0 | 2549.88 | 2588.34 | 328.97 | 337.96 | 2.74 |

and H4) are marked by red arrows. All strong IRHs can be traced smoothly across the transect. This supports our point that those data are well combinable.

The horizontal resolution of the bed reflection is best in the CReSIS data. Whereas in the other data the bed reflection is somewhat blurred, we get a well-defined bedrock topography from the CReSIS data. However, these quality differences could easily be induced solely by the different processing techniques, or to some extent influenced by the different locations of the profiles. Thus a comparison at this stage is of little help for judging the actual systems' capabilities in depicting the bed topography. If a more quantitative comparison of this aspect is required, we propose a survey with all systems measuring a common profile with a length of five to ten ice thicknesses.

### 6.1.2 Differences in IRH depth and strength

Other reasons for the reflector dissimilarities is that the measurements were not conducted at exactly the same location, with different measuring frequencies and the measurements have different path orientations as well. The depth of an isochronous IRH is dependent on surface mass balance, ice thickness and ice flow. Urbini et al. (2008) and Frezzotti et al. (2005) found significant spatial variability in snow accumulation on the scale of a few kilometers around Dome C (2% within 2 km in direction SW–NE, inferred from Urbini et al., 2008, Fig. 7b). This can have the effect that some layers may be thicker in one profile than in another, leading to different appearances of the according reflections in the radargrams. It is also possible that some signals are missing completely at one location, due to erosion processes (Frezzotti et al., 2005). When stationary, even small spatial accumulation variations cause spatial variations in reflector depths. This is certainly the case for greater depths, where the effect accumulates over thousands of years. We see the reflector-depth variations as TWT shifts of identified reflections in the traces in Fig. 2 and

in the slopes of IRHs in Fig. 3. For instance the reflector at 11 $\mu$s in the CReSIS data has a slope of about 0.1 $\mu$s km$^{-1}$, corresponding to about 8.4 m km$^{-1}$. For the reflector at 26 $\mu$s in the same data the slope is 0.15 $\mu$s km$^{-1}$ or 12.6 m km$^{-1}$. The spatial variations are thus big enough to explain the differences in TWT for the identified IRHs in the different RES data, e.g. about 0.1 $\mu$s TWT shifts for reflectors in the CReSIS profile, which is more than 1.5 km apart from the other data.

Since ice thickness is a factor for IRH depths, the slopes of IRHs are also influenced by the bed topography. We find quite steep slopes in the bed reflections in Fig. 3. The ice thickness varies significantly, even on the scale of the distances between the RES profiles. The bed reflection varies about 1.0 $\mu$s TWT over 1 km profile length at the steepest slopes of the bed reflections. This variation corresponds to approximately 80 m km$^{-1}$ in bed elevation change. This is consistent with results by Rémy and Tabacco (2000), who established a 50 km × 50 km bedrock map for the Dome C region with 1 km horizontal resolution. They found valleys, a few tens of meters deep, close to the Dome C drill site. So, again, the differences in the RES data, e.g. the 0.5 $\mu$s earlier bed reflection or earlier IRH reflections in the CReSIS trace (Fig. 2), can be explained by their spatial separation. Furthermore do the RES profiles have different orientations. This could be the reason for some reflectors, fully or partly induced by COF changes, to be weaker or not existent in some of the data, as the power, reflected by those horizons is dependent on the electric-polarization direction (Matsuoka et al., 2003; Eisen et al., 2007). Another factor that is worthwhile considering are the different measuring frequencies of the RES systems, the center frequencies of which range from 60 MHz to 195 MHz. The reflection coefficient of conductivity-caused horizons is frequency dependent (Fujita and Mae, 1994). This leads to different reflection amplitudes or reflector strength of one certain horizon in radar data measured with different frequencies, in case the

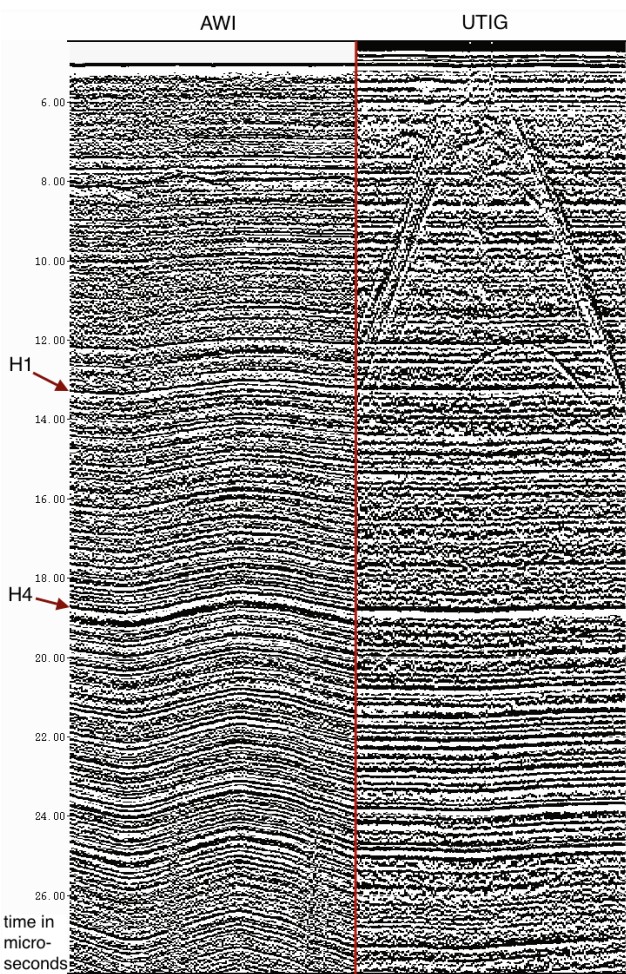

**Figure 4.** Intersection of AWI and UTIG profiles about 1 km northeast of the drill site (see Fig. 1) for direct visual comparison. Two examples of identified IRHs are indicated by red arrows. Note that the air-ice reflection is shifted to 5 $\mu$s. All strong horizons can be traced smoothly across the intersection.

corrected relative returned power (in dB) is plotted. According to MacGregor et al. (2015b) also the conductivity itself is slightly frequency dependent. This causes a signal to be more or less attenuated, depending on its frequency. This does not influence the TWTs or inferred depths of IRHs and thus is not so relevant for studies of the age structure of the ice sheets. However, the frequency dependence of conductivity certainly is a factor that should be closely examined when using RES data to deduce attenuation rates and ice temperature.

### 6.1.3 Penetration depth

The IRHs cease to be visible at different depths in the bottom section of Fig. 3. This happens in a different manner for the different RES data. In the AWI and INGV data it is a rather slow process, with IRHs gradually becoming weaker. The weaker IRH response is due to the attenuation of the RES signal as it propagates through the ice. As such, weak internal reflectors are difficult to detect, whereas the strong return from the ice–bedrock interface can still be detected. In contrast, the UTIG, CReSIS and BAS data clearly show reflectors down to approximately 33 $\mu$s, with comparably strong "last" reflectors (green arrows) and below that depth no continuous IRHs are visible. The lack of continuous IRHs below is not an issue of the systems' power, but rather means that the former horizons are in some way deformed, amalgamated, or disrupted into small scale structures, not resolvable by the radar systems. This basal region, indicated by a sudden drop of returned power in the radar data is described as echo-free zone (EFZ) by Fujita et al. (1999) and Drews et al. (2009). The lack of IRHs in this range not being an issue of the systems' power is also supported by the fact that there are regions with to some extent spatially coherent signals almost down to the bedrock, like in the BAS data and also in the UTIG and CReSIS profiles, outside the 5 km range shown in Fig. 3. It is difficult to give a comparing judgement on the quality of the three RES systems in this deepest part. The profiles were not measured at the same location and the internal structure in the ice close to the bed differs strongly. But as we still see some small scale structures and coherent signals, we are confident that all three systems are able to image structures in the basal layer reliably. As the basal layer comprises the old ice, this is a crucial factor for Oldest Ice reconnaissance surveys. The lower range resolution of the BAS system, comparative to the other two systems, applies also for the basal layer.

### 6.2 Synchronization of RES and ice core data

We find that distinct patterns of IRHs in the RES data can also be found in the synthetic trace. The pronounced reflectors that are identified in all RES data can also be matched with the synthetic data. In this way their depth origin at the drill site and thus their age can be determined, as described in Sect. 5.3. However, like for the measured RES data, there are also pronounced differences in appearance between the RES and synthetic data. Possible reasons are discussed in Sect. 6.2.1. The uncertainties on depth and age are discussed in Sect. 6.2.2.

### 6.2.1 Differences between modeled and measured data

The dissimilarities in the appearance of reflectors described for the comparison of the different RES data are even stronger when comparing any of the measured RES data with the synthetic trace. This is consistent with the explanation by different measuring sites and spatial variability in accumulation and ice thickness. The synthetic trace is calculated from the ice core data. The EDC99 core and the RES profiles are a few hundred to two thousand meters apart. Additionally, the core, with its 0.1 m diameter, samples a much smaller area than a radar system that averages over a footprint on the

100 m scale. There were discussions about the spatial significance of ice core signals (e.g., Fisher et al., 1985; Richardson and Holmlund, 1999; Veen and Bolzan, 1999). Palais et al. (1982) and Münch et al. (2016) examine the representativity of cores, using snow accumulation variability and stable-water isotope variations, respectively. Frezzotti et al. (2005), too, get slightly different accumulation rates from ice core and radar measurements and explain them with the different sample area. However, they find that for the Dome C region the differences of core and radar measurements of 3% and the spatial variability of accumulation are relatively small compared to other East Antarctic regions, and the smallest of their Dome C–Terra Nova Bay traverse.

Wolff et al. (2005) compare the conductivity record of the EDC99 core with the one of the first drilling attempt (EDC96) that is 10 m away. They find that only in 45% of the cases the largest conductivity peak in a 10 m section is also the largest peak in a 10 m section of the parallel core. This is typical of low accumulation sites, because significant parts of a single year's signal can be lost at a location by post-depositional processes like snow drift. Gautier et al. (2016) evaluate the variability of the volcanic signal at the EDC site, using five 100 m firn cores, drilled 1 m apart from each other. They found that the probability of missing volcanic events is 30% when using only a single core. The Tambora event (1815 CE), for instance, is not detected by their algorithm in two out of five cores. Wolff et al. (2005) suggest to use methods that sample a larger volume of ice to smooth out the spatial inhomogeneity. RES being such a method explains that the RES data are more similar among each other than RES versus synthetic data. This implies also that a very strong RES reflector does not necessarily have to be a large peak in the conductivity profile or synthetic trace.

For some quantitative comparison of amplitudes, e.g., for deriving the attenuation rate, again, the different source wavelets have to be taken into account. The synthetic trace is calculated with a monochromatic burst wavelet. We did neither scale reflection amplitudes to the differing center frequencies of the CReSIS and UTIG systems nor account for the finite bandwidths of the chirp systems. Furthermore, conductivity itself is frequency dependent (MacGregor et al., 2015b). This implies that also the different measuring frequency of the DEP compared to the source wavelet (100 kHz vs. 150 MHz) effects the reflection amplitudes of the synthetic trace. However, the uncalibrated conductivity profile of the Dome C core and the simple model itself do not allow to quantitatively analyze the reflection amplitudes, but only the signature of reflection patterns in our study. In addition to these factors, all related to the intrinsic frequency dependence of the dielectric properties, also the frequency dependent thin-film interference is influencing the measured and modeled amplitudes. The signals from interfaces between layers with constant dielectric properties can be strengthened or weakened by constructive or destructive interference, depending on the thickness of the layers and the source frequency. This may complicate any data combination that is dependent upon amplitudes yet further.

### 6.2.2 Uncertainty assessment

Uncertainties incorporated in the synthetic data are the uncertainties from the ice core measurements, inaccuracies in determining the permittivity and the neglect of temperature and anisotropy. The uncertainties on the density decrease with depth, due to higher absolute densities and Hörhold et al. (2011) give a value of 0.66% at 100 m depth. The errors in the permittivity, induced by neglecting the complex character in the air-ice mixture, are negligible, as we exclude the firn section and look only at depths greater than 600 m, where the air partition is minor. Uncertainties on wave velocity due to the dependence of the permittivity on temperature and anisotropy both stay below 1% (Gough, 1972; Matsuoka et al., 1997; Fujita et al., 2000).

The errors in the synthetic trace have an influence only on the TWT of the synthetic reflectors, and thus eventually on matching those with the RES reflectors, but not on the actual depth and age assignment. Sensitivity studies with the conductivity profile define the depth range in which a synthetic reflection peak has its origin. Thus the depth uncertainties of the IRHs are given by the depth uncertainties of the DEP measurements and the vertical resolution of IRHs in the synthetic trace. The latter is adjustable, as it is defined by the bandwidth of the source wavelet and the smoothing filter, as shown by e.g., Cavitte et al. (2016). The depth uncertainties of the DEP measurement are reported as about 2 mm by Wolff et al. (1999). Added to that comes the depth uncertainty of the ice core itself, which is difficult to quantify. Due to breaks in an ice core, an inclined borehole, and core relaxation after drilling the logged ice core length is always different from the true depth. Parrenin et al. (2012, Section 2.2.1) estimate the offset to reach up to several meters for a deep drilling. The direct transfer of reflector depths by the sensitivity studies is not completely true for the RES reflectors. Because of different bandwidths used in the RES data, there is an uncertainty associated in matching the peaks between the synthetic and RES data sets. All of the BAS and INGV reflections, for example, are wider than the ones in the synthetic trace and thus it is not clear which conductivity peaks are incorporated in one RES reflector. The age uncertainties due to reflector width increase with depth and can be inferred from Fig. 5, which shows the gradient of age with depth (blue curve), and with TWT (black), respectively. Added to that comes the age uncertainties from the AICC2012 age scale itself, given in Table 2 (Veres et al., 2013; Bazin et al., 2013). The curves of Fig. 5 also give the magnitude of error in age that is avoided by our method compared to using only a RES profile, in some distance to the core, and the age scale of the core. For example, the RES profile is 1 km away from the core and the IRH of interest has a slope of 10 m km$^{-1}$ in the direction drill site–RES profile. When using only the

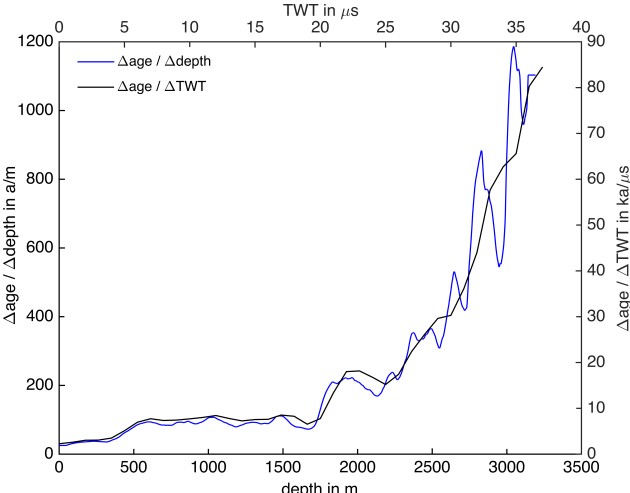

**Figure 5.** Gradient of the age from the AICC2012 age scale with depth (blue, left and bottom axes) and TWT (black, right and top axes) to infer age uncertainties due to reflector width and slope.

RES profile, we would assign the reflector with an age off by 10 m on the ice core time scale. For a reflector at about 10 $\mu$s TWT that would correspond to an age shift of about 1 ka, for a reflector at 32 $\mu$s of about 8 ka. This shows the advantage of our approach of first matching the RES IRHs with the synthetic radargram and only then determining the depth and age of the reflectors at the ice core location, as it eliminates one possible source of error. The extent of error reduction, however, does depend on IRH slopes and distances of the RES profiles to the ice core site.

## 7 Conclusions

In our study we compared the data sets of five different RES systems, addressing the questions of their compatibility for combined usage and suitability for informing potential "Oldest Ice" site characterization. All RES profiles were recorded within a 2 km radius around the EPICA Dome C drill site, where the current oldest ice sample was retrieved. We found that the data are broadly comparable at that location and that the most-pronounced reflectors can be found in the RES data. The main differences between the RES systems are constituted by their resolution of englacial structure and bedrock and their quality in imaging the basal layer. The CReSIS data have the best horizontal resolution in the depth of the bed and are thus providing a well-defined subglacial bedrock topography. At this stage it is inconclusive, if this can be attributed to the CReSIS system, or rather to the different processing technique and profile location. If interested in vertically well resolved IRHs at intermediate depths, the AWI, CReSIS, and UTIG systems, are the most suitable, due to their comparably high vertical resolution. However, we did not investigate the continuity of the IRHs beyond the 5 km profile lengths. Based on their close similarity in reflection patterns at the investigated location we assume that the AWI, CReSIS and

UTIG data are smoothly combinable for common interpretation, which is supported by the direct comparison of AWI and UTIG radargrams at their profiles' cross-over point. However, the even higher vertical resolution of the CReSIS data might cause some difficulties for the transfer of certain IRHs, as they might transition into multiple peaks. But this has to be checked directly at the cross-over points for the IRHs of interest. The CReSIS, UTIG and BAS systems have the largest penetration depth and are able to image some structures in the basal region. Nevertheless, due to the comparably low vertical resolution of the BAS data, we attest the CReSIS and UTIG systems the best overall suitability in our comparison for Oldest Ice reconnaissance surveys. The AWI and INGV data in the current version are not as convenient for this purpose, as they fail to depict the internal structure in the deepest approximately third of the ice thickness at EDC. Nevertheless, the profiles could be used to close data gaps with respect to IRHs at intermediate depths and ice thickness.

In addition to the comparison of the RES data, we synchronized the measured RES data with a synthetic radar trace for depth conversion. Input for the forward model for the synthetic trace were the EDC ice core conductivity and density. We found that the RES data are more similar among each other than compared to the synthetic trace. This can be explained by the spatial variability of the strengths of single conductivity signals as sampled by ice cores and the smoothing effect of RES measurements due to their larger footprint and lower vertical resolution. Another reason for the differences is the partly differing frequencies and source types of synthetic and RES data, which have an influence on the reflection amplitudes. This factor needs to be examined more closely for quantitative amplitude analysis, which is necessary for inferring the attenuation rate and ice temperature from IRHs. Such analysis, however, is not feasible with the data available for this study, as the DEP measurements do not allow for correctly reproducing reflection amplitudes. Despite the differences we were able to match 10 pronounced reflectors in the RES and synthetic data. We identified the causative conductivity peaks of the matched IRHs and in this way determined their depth and age. Since the identified IRHs are conductivity-caused and thus isochronous they can be used to extend the age structure, provided by the Dome C ice core, to regions of interest for an Oldest Ice drill site.

The combining of different RES data and the dating of horizons is requisite for a large-scale mapping of the age structure of the East Antarctic ice sheet, where no reasonable coverage and resolution is obtained with only one data set. The age architecture in turn facilitates the inference of spatial and temporal variations of mass balance and provides boundary conditions or parameters for large scale ice-flow models.

*Acknowledgements.* Operational support was provided by the U. S. Antarctic Program, the Institut Polaire Français Paul- Emile Victor (IPEV), the Italian Antarctic Program (PNRA and ENEA), and the Japanese National Institute of Polar Research (NIPR). We acknowledge that several CReSIS faculty, staff and students contributed to the development of radars and antenna arrays used to collect data reported here. The UTIG line was funded as part of NSF's Interna-

tional Polar Year activities (grant ANT-0733025) to the University of Texas at Austin, and the UK's NERC grant NE/D003733/1 to University of Edinburgh. Additional support was provided by the French ANR Dome C project (ANR-07-BLAN-0125). We would like to thank T. Jordan and one anonymous reviewer for their comments which greatly improved the original manuscript, and R. Bingham for handling our submission. This is UTIG contribution ####.

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
