# Peer review of "Comparison of measurements from different radio-echo sounding systems and synchronization with the ice core at Dome C, Antarctica"

_The Cryosphere, 2016_

## Referee Comment (RC1) · T. Jordan (Referee) · 11 Jul 2016

**Review of: Radio-echo sounding measurements and ice-core synchronization at Dome C, Antarctica**

Anna Winter et al. 2016, *The Cryosphere.*
Reviewer: Tom Jordan, University of Bristol, 11th July 2016.

**Summary**

The manuscript provides a detailed compatibility study of RES data from five different radar systems in the Dome C region, and is focused around the detection of Internal Reflection Horizons (IRHs). Their primary glaciological motivation, which is well emphasised throughout, regards synchronisation with ice core data and Oldest Ice characterisation. They present comparative A-scope and Z-scope plots, and are able to relate the different features present to the differing radar system characteristics. As part of the study they use an electromagnetic modelling framework and the dielectric profile of the ice core to estimate a `synthetic' radar trace. The synthetic trace is a valuable component of the investigation as it enables them to establish a causal link between the dielectric properties of the ice layers and the internal reflections present in the radar traces. Detailed discussion of relevant uncertainty and spatial variability is provided. The authors conclude that the AWI, UTIG and CReSIS systems provide the best resolved internal layers, and have the best potential to be combinable. Additionally, the synthetic trace/age conversion enables the authors to conclude ~10 IRHs can be well synchronised with the ice core timescale.

**General comments**

Overall the scientific analysis is of a high quality and the manuscript is well written. I have, however, made a few suggestions where more detail and precision in the presentation is required; particularly in the methods section. The manuscript is well structured and referenced, with informative figures and tables. Regarding the novelty of the study, I think it needs to be made more explicit as to what differentiates the manuscript from Cavitte et al. 2016 (which also considers IRH detection at Dome C for different radar systems). I appreciate that there are differences, (e.g. the use of the synthetic trace in this study), but this may not be obvious to the general reader.

The electromagnetic modelling framework for the synthetic trace appears well established, and physically rigorous. However, given the overall emphasis on comparing how radar system characteristics influence the sounding data, one major area which requires explicit investigation is frequency dependence (see specific comments, Sect. 2.2, Sect. 5.1). Whilst probably not directly impactful for Oldest Ice/synchronization, frequency dependence could be important for combining IRH data sets where reflection amplitudes are important, and thus is required to complete the overall compatibility aspect of this study.

The manuscript is clearly of general interest to the readership of *The Cryosphere*. RES of IRHs can provide useful glaciological information that goes substantially beyond ice core chronologies; for example temperature (from attenuation) and ice dynamics (from IRH derived metrics such as the continuity index). Such a comprehensive compatibility study between different radar systems is therefore of suitably high impact, and will no doubt be a central reference point for future studies which combine RES data sets. Whilst I appreciate that the glaciological application here is Oldest Ice, I have made a few specific comments regarding how the authors could broaden the scope of their introduction and discussion.

**Specific comments**

**1.**

A clear case needs to be made in the introduction what differentiates the study (both in terms of the questions addressed and the methods that are used) from Cavitte et al. 2016.

As mentioned in the general comments, I think the emphasis in the introduction on Oldest Ice is too narrow. This is an excellent opportunity to communicate to the wider glaciological community what rich information is present within RES data (in particular derivable from IRHs), and therefore could be exploited on a broad scale if different data sets can be combined. Two relevant examples are: depth-averaged temperature (from the attenuation rate inferred from internal reflections), Matsuoka et al. 2010, Macgregor et al. 2015b, ice dynamics (from the IRH continuity index) Karlsson et al. 2012.

**2.1**

It would be helpful here to provide more information regarding the DEP method and the gamma absorption method. In particular; how some of the underlying assumptions/restrictions of these methods could impact the rest of the investigation. For example, in the case of the DEP method, I think that it is important to note explicitly that the reference frequency (100 kHz) is significantly less than the radar systems (~100 MHz), which may be important regarding frequency dependence of dielectric conductivity/attenuation (see recent discussion in Macgregor et al. 2015b)

**2.2**

Given the range of radar system frequencies that are considered in this study (60-195 MHz) I think it is necessary to investigate the frequency dependence of the synthetic trace. Note; that this is in the context of thin-film interference (and how frequency/wavelength dependence affects the peaks/relative amplitudes of the synthetic trace) rather than the intrinsic frequency dependence of the dielectric conductivity as mentioned above. Specifically, it would be useful to repeat the synthetic trace analysis, for source wavelets at the CReSIS (195 MHz) and UTIG (60 MHz) centre frequencies. If pronounced sensitivity is demonstrated, then these repeat traces could be used to improve the comparison between the synthetic A-scope trace and the radar A-scope traces in Section 5 and the related discussion in Section 6. I hope, given the excellent EM simulation framework available to the authors, this is request is fairly straightforward to do, and would add a significant value to their investigation.

On a related note, I assume that the EM simulations assume a monochromatic source? Since the chirped radar systems have both finite and differing bandwidths, this is clearly an important simplification. Finally, is it also correct that the EM simulation method is physically a closer representation of the pulse/burst systems rather than the chirped radar systems? Again this needs to be discussed, along with potential caveats for cross-comparison.

**2.3**

Given the overall emphasis on cross-comparison between different radar systems, it is desirable that the authors provide evidence for how robust the method used to determine the permittivity of ice is for other radar systems (only AWI is discussed here). The variance in the obtained values could then be discussed in relation to other components of their investigation.

**5.1**

If possible I would like to see a discussion (and potentially an explanation) for the variation in the relative amplitude of the IRH peaks for the different radar systems. In particular, the relative amplitude of the peaks for the UTIG trace appears to be lower than the other systems of comparable vertical resolution (AWI,CReSIS), and this may potentially relate to frequency dependence. One reason why reflection amplitudes are important is that they are used to determine depth-averaged attention rates (and thus information about depth-averaged temperature). Subsequently, even if only preliminary conclusions be made regarding differences in amplitude, this will be useful for future combination studies.

Since it is mentioned in the conclusions that AWI, CReSIS and UTIG are likely suitable for combined analysis, it would be useful to see some direct cross-over analysis of the traces. I appreciate from Fig 1 that this may only be possible for AWI and UTIG, but this would act to strengthen the overall conclusions regarding combining data.

**6.1.1**

I think it would be helpful here (or potentially in the caption of table 1) to provide relevant equations regarding the relationship between bandwidth and vertical resolution for the chirped systems (e.g. as supplied in the CReSIS reference).

**6.2.1**

Is it possible to provide a rough estimate of the radar footprint diameter in the Dome C region? My guess is that as we are dealing with comparatively thick ice this would be toward the upper end (or possibly exceed) the range that is stated.

**7.**

As with the introduction, I think a clear case needs to be made what distinguishes the conclusions of this study from Cavitte et al. 2016.

I think it also needs to be discussed explicitly in the conclusion how variable vertical resolution (particularly how multiple peaks transition to single peaks as function of vertical resolution), pose challenges for combining data sets. This discussion will hopefully also address other glaciological information derivable from IRHs.

**Minor comments, typographical errors, etc.**

Note; I use the symbol `→' to indicate my suggestions for ` replace with'.

**General**

There are quite a few examples where there is no white space preceding the SI units. (e.g. 0.2m). These should be corrected.

Be consistent with the hyphenation of ice-core/ice core

There are many instances where use of `vertical resolution' would be less ambiguous than `resolution'.

**1.**

Line 17: It would be helpful to add a reference here.

For brevity, the final paragraph of the introduction could be dropped.

**2.1**

Line 1: ice: → ice (: should only be used for equation arrays/lists)

Line 9: Reword `we shortly discuss the input parameter permittivity of ice.

Lines 11-13: If possible, please reference the core data (temperature, accumulation, etc.)

Lines 18,22: Give units for sigma and rho when they are introduced.

Lines 24,27: Unit spacing for Xmm

Line 24: measuring: → measurement

Line 25: comma before rho_ice

**2.2**

Title: Consider changing to `Electromagnetic modeling of radar traces'

Line 1: ice: → ice (: should only be used for equation arrays/lists)

Line 4/equation 2: The equation is correct, but the symbols (epsilon,epsilon',epsilon'',sigma, omega ) must be introduced correctly in text; see, for example, Eisen, et al. 2004, equation (1). Additionally, episilon_0 is best described as the `vacuum permittivity' (the use of ordinary is confusing since `ordinary permittivity' is used in the context of anisotropic media).

Line 14: `incorrect ordinary permittivities (the real part of the complex relative permittivity)' → `incorrect real permittivities'.

Line 24: Measuring → measurement

Line 17. '1D-FD' → '1D-FD (One-Dimensional Finite Difference).'

Line 19: Please be more specific about the boundary condition(s). Is it the lower or the side boundaries?

Line 20. Please reference the Courant Criterion, and state what it tests for (convergence of the numerical solution).

Line 21. See my specific comment about the frequency dependence/sensitivity of the synthetic trace. It should be clearly started here that there are differing frequencies and bandwidths for the radar systems.

Line 24. Please provide a reference for Hilbert magnitude transform.

**2.3**

Title: Consider changing to `Determination of the relative permittivity of ice'

**3.2**

Line 17,22: Unit spacing for X microseconds.

**3.4**

Line 27: Unit spacing for km.

**3.5**

Line 27: Unit spacing for MHz.

For completeness it would be helpful to list the distances of the profiles from Dome C for all radar systems (this is only provided for AWI and CReSIS).

Line 15: No new paragraph?

**5.1**

Line 12: second → upper right

Line 18: resolution → vertical resolution

**5.2**

Line 6: y → vertical

Line 8: about → an approximately

Line 17: is starting → starts?

Line 32: missing → missing from the synthetic trace

**5.3**

Line 2: Is there a suitable reference for the `sensitivity approach' used here?

Line 15: The advantage → The advantage of the sensitivity approach…?

**6.1.1.**

Title: Consider changing to `Vertical and horizontal resolution'

Line 14: UTIG systems the→ UTIG systems to be the?

Line 15: Obviously in → Due to their lower vertical resolution

Line 25: continue → continue with

**6.1.2**

Line 3: remove `and others'

Line 8: accumulated → accumulates?

Line 10: it is → the slope is

Line 13: Urbini et al. needs a date.

Line 14/15: Reword sentence starting: `That would…

Line 22: they → this

**6.1.3**

Line 8: comparing → comparative

**6.2**

Lines 12-14: 6.2.2. and 6.2.1 are introduced in the wrong order.

**6.2.1**

Lines 19,20,28: Unit spacing for Xm

Line 23: examine → who examine

Line 28: find → found

**6.2.2**

Line 14: missing full stop after trace

Line 15: remove `used'?

Line 27: example: The → example, the

**7.**

Line 9/10: It probably best to relate the well resolved IHRs for these systems directly to their better vertical resolution than the other systems (rather than implicitly through their bandwidth).

Line 13: Reword: The best quality in imaging the basal layer have the CReSIS, UTIG and BAS data, the latter, however, with …

Line 14: Reword

Line 19: Remove `profound' (it is best practice to avoid superlatives)

For the conclusions it is best practice to use past tense rather than present tense, and I would recommend carefully checking this section.

**Tables, figures and captions**

The tables and figures are both informative and well presented. I do, however, have a few suggestions.

**Table 1**

Relabel `resolution' as `vertical resolution'. For completeness it would be desirable to provide an indication of how windowing/processing affects the vertical resolution (e.g. for the CReSIS system I believe that the post-windowed vertical resolution is ~4.3 m).

**Fig 2.**

Given that the reflections are ultimately caused by discontinuities in conductivity (rather than the peaks themselves), I think it would be useful to provide a plot of the vertical gradient of conductivity underneath the conductivity plot. I wouldn't be surprised if this gradient plot has a more `immediate' correspondence with the synthetic trace reflections, and could be used to improve the analysis in Section 5 and 6.

Following the terminology in Section 4, it should be added explicitly to the caption that Fig. 2 is an A-scope plot

**Fig 3.**

The font size for the axes labels/numbers should be increased

of → for

x-axis → horizontal axis

closer →closely?

**References**

Cavitte, M. G., Blankenship, D. D., Young, D. A., Schroeder, D. M., Parrenin, F., Lemeur, E., Macgregor, J. A., and Siegert, M. J.: Deep radiostratigraphy of the East Antarctic plateau: connecting the Dome C and Vostok ice core sites, Journal of Glaciology, 2016.

Eisen, O., Nixdorf, U., Wilhelms, F., and Miller, H.: Age estimates of isochronous reflection horizons by combining ice core, survey, and synthetic radar data, Journal of Geophysical Research: Solid Earth (1978–2012), 109, doi:10.1029/2003JB002858, 2004.

Karlsson, N. B., Rippin, D. M., Bingham, R. G., and Vaughan, D. G.: A 'continuity-index' for assessing ice-sheet dynamics from radarsounded internal layers, Earth and Planetary Science Letters, 335, 88–94, doi:10.1016/j.epsl.2012.04.034, 2012.

Macgregor, J. A., Li, J., Paden, J. D., Catania, G. A., and Clow, G. D.: Radar attenuation and temperature within the Greenland Ice Sheet, Journal of Geophysical Research: Earth Surface, 120, 983–1008, doi:10.1002/2014JF003418, 2015b.7/jog.2016.11, 2015b.

Matsuoka, K., Morse, D., and Raymond, C. F.: Estimating englacial radar attenuation using depth profilesof the returned power, central West Antarctica, Journal of Geophysical Research, 115, 1–15, doi:10.1029/2009JF001496, 2010.

---

## Referee Comment (RC2) · Anonymous Referee #2 · 27 Jul 2016

This manuscript presents a first comparison of radar data collected with five different systems in the vicinity of Dome C. All of these airborne radar systems (AWI, UTIG, CReSIS and BAS) have generated the majority of the radar data in Antarctica and Greenland so that I see an overall merit to compare these datasets, not only for the oldest ice site survey, but for the ice-sheet research in general. However, the comparison presented in this manuscript is not at all rigorous. It is more or less just a visual inspection to develop fuzzy impressions (that many people already have, I believe), rather than a careful, scientific comparison to rigorously see what can be said and what should not be said by synthesizing different radar datasets together. The goal of the analysis is to compare the RES and synthetic radar data in terms of identifying distinct reflectors that can be found in all datasets and that can be confidently be matched in between the different datasets (P6L10-12). The analysis presented in this paper is

inadequate to make this point. As I point out below, I see many not-well-justified procedures in the data/method section. Also, relevant information are shown in many places in the paper, so it is very hard to develop a confident understanding on the analysis presented here.

Individual comments (there are editorial comments as well, but only a few):

Title: With this title, readers cannot find that there is the first comparison of the radar data collected with five different systems. How about "Comparisons of radar data collected by different systems to synthesized radar data using the Dome C ice core, Antarctica"?

P1L3: bedrock -> bed, bed can be sediment, and not always rock.

P1L4: quality -> capacity?

P1L10-13: please improve the manuscript, otherwise the statements here are not well supported.

P1L13: Add "EDC's" before AICC2012. Is it necessary to be so specific on the timescale in the abstract?

P1L17: perfect -> valuable. Nothing is perfect.

P1L18: "air bubbles and hydrates"

P2L3: revise "this new, older core". The oldest core is not drilled yet.

P2L8: I understand that age structure refers three-dimensional age distribution within the ice sheet. However, it is not necessary to identify the oldest ice. Please clarify what is needed for the oldest ice survey and separately for more general interests of the ice-sheet research.

P2L9: delete "dielectric properties" in front of "density".

P2L10-17: Fujita and Mae (1994, Ann. Glaciol.) is the first paper to present the frequency dependence of reflectivity in the ice. In the East Antarctic inland, the major reflection cause can be acidity or COF, depending on the radar frequency, ice temperature, and acidity/COF contrasts (Fujita et al., 1999 in author's reference list). In general, acidity-based reflection is more dominant at lower frequencies than 50-100 MHz, and COF-based reflection is more dominant at higher frequencies than 100 MHz. Such radar frequency dependence should be briefly mentioned here. And in a later section, the authors should address how 60 MHz data (UTIG) and 150-200 MHz data (other systems) can be compared, even if reflection causes are not necessarily identical.

P2L16: COF-based reflections do not necessarily constitute isochrones by definition, but Fujita et al. (1999) argued that COF contrasts can possibly be initiated by acidity contrasts so that regardless of the reflection cause IRH detected at any frequencies can be used as isochrones. This view is supported by a wide range of agreements between modeled isochrones and IRH observed at different radar frequencies. This work also supports this view.

P2L19: change to "from any ice core, if the isochrones. . ."

P3L7: "In the sections below we describe, the ICE-CORE data used for. . ."

P3L11-14: Please add adequate references to characterize EDC core sites. I don't think that Augustin et al. (2004) alone shows the full range of information presented here.

P3L19: If I understand correctly, ice temperature is assumed to be -15oC uniformly throughout the core. It is not the case. If the authors just need to have conductivity contrasts to identify acidity-based IRH depths, please say so clearly here to justify the uniform temperature assumption. Anyway, I cannot really understand the motivation of quite complicated (CPU expensive) modeling under such extremely simple assumption.

P3L20: why can the conductivity at the surface be assumed as 4.05 micro S/m? This

interpolation is made only for about 7 m so it does not make any major difference. However, I cannot see a reason why the conductivity at the surface has this value. If it is not well justified, why don't you assume it constant over this 7-m-long segment?

P3L27: add space to 5mm.

P4L2: Is Equation (2) necessary?

P4L8: permittivity is assumed to be 3.17; any reference? Is it reasonable for the temperature range measured at Dome C? Please present an ice temperature profile measured in the borehole.

P4L10-11: I don't agree. IRH is a result of many individual reflections caused at conductivity contrasts located close to each other. To calculate such interference of many reflected waves, phase is important and phase is dependent on conductivity (its value, not only the contrasts). If author's argument is really the case (i.e. only conductivity contrast is necessary), the authors can simply use the DEP results without any modeling. The bottom line: please clarify what "for the purpose of reproducing reflections" really mean.

P4L11-13: hard to understand; please revise. Radar data are collected in the two-way travel time domain. Do authors want to say "reflections in the depth domaion" (not the TWT domain)?

P4L14: Add space, 0.2 m

P4L14-15: It is a reasonable approach, but explicitly say that in this way only conductivity-based reflections are modeled, and permittivity-based (i.e. COF and density based) reflections are not modeled (Fujita and Mae, 1994, AGlac).

P4L19: model's depth/time increments are interchangeable in this context, I believe. 0.02 nsec and 20 mm are not equivalent if the propagation speed is for e=3.17.

P4L21: What's the exact purpose to use EMICE? Ice temperature is assumed uniform.

Permittivity is smoothed out. Input radar waveform to EMICE is inconsistent with any of the radar system. Is it really necessary to use EMICE? Why is the model output smoothed over 150 nsec (equivalent to about 13 m in ice)? If so, please use the model for more realistic conditions.

P4L27-31: This is a major tuning. It is assumed that AWI data (not exactly at the core site) and synthesized results show consistent englacial reflections and permittivity is tuned. Such assumption should be mentioned more explicitly. What is the corresponding ice temperature to e=3.17? What is the range of permittivity along the core associated with the ice temperature variations? Overall is e=3.17 a reasonable assumption here?

P4L32: which IRH are compared between AWI's data and synthesized results? All IRH?? Revise "For this value, the identified reflections occur at the same TWT for both traces". Again, permittivity is temperature dependent. And, somewhere further down in the manuscript, firn correction of 10 m is made (Fig. 3 caption). Then I am really puzzled; information necessary to understand the TWT/depth conversion scattered many places in this paper so it is really hard for me to follow author's logic.

P5L8: "(CReSIS) at the University of Kansas, . . ."

P5L11ff: Please use Table 1 more effectively. The range of information given in the text is also given in the table. References should be added to identify the processing procedure; descriptions here are too vague and brief to have the full understanding of individual datasets. In addition to items currently presented in Table 1, it is useful to show flight height, stacked distance, reference to processing procedure, etc.

P5L19: What does "unprocessed" exactly mean?

P5L28: What is "pulse envelope radar"? Is it a pulse-modulated radar that records only the returned power, not phase?

P6L6: What is an automatic gain control? I guess that it is a way to compensate

geometric spreading and attenuation within the ice, but no details are given. Is this gain adjusted correctly so that the gain is not increased once the received power reaches to the noise level?

P6L10-12: Here, the goal of the analysis is clearly articulated: "our aim is to compare the RES and synthetic radar data in terms of identifying distinct reflectors that can be found in all datasets and that can be confidently be matched in between the different datasets." However, as I pointed out above and will do so below, the analysis presented here is not adequate to meet this goal.

P6L24: Physics behind the sentence "The exponential trend is removed from every trace" is that (1) ice temperature is uniform from the surface to depths, (2) chemistry is also inform, and consequently (3) attenuation rate is uniform. This feature may be seen in the model results, but if so it is only because the ice temperature is assumed to be uniform in the model. Again, I am really puzzled; what do the authors want to replicate by the model and for that goal what can be simplified? Information on these points are scattered many places in the manuscript so it is very hard to read.

P6L26-27: Please revise. I cannot understand. Figure 2 shows the returned power in arbitrary scale; is it linear scale or dB scale? If the latter is the case, does the panel show log of log??

P7L1: remove approximately. One third is approximate anyway.

P7L10-11: repeated/duplicated information. Delete.

P7L13-14: I cannot agree at all with the authors. Not all of ten IRH are confidently matched correctly. My quick view found no H6/H10 in AWI, no H3/H6/H9/H10 in UTIG, no H2/H3/H6/H9 in CReSIS, no H1-H6/H10 in INGV, and no H3/H6/H9 in BAS. I don't expect that all of these features match pretty well between the all datasets. This level of agreement is something I don't surprise and it is indeed a new finding of this analysis. Please articulate what you found; don't stretch your results. I did not comment the

rest of Section 5.1 and most of Section 5.2; please revise it accordingly based on my suggestions above.

P8L25: the term "echo-free zone" is misused here, if the authors follow the original definition made by Fujita et al. (1999, JGR). It does not simply refer the ice from which no echo is received. It's upper surface is associated with a significant, sharp drop of the returned power, indicating the sudden loss of reflection even if the incident radio wave is strong enough (not attenuated so much).

P9L2-4: Again I'm really puzzled. What is the sensitivity test? AWI data were already used to have a best estimate permittivity/propagation speed. Why is conductivity mentioned here, though it is irrelevant to depth estimate? I did not comment on the rest of Section 5.3.

P9L29-30: Please make more rigorous discussion. For me, Sections 5.1 and 5.2 are inadequate to draw this conclusion.

P9L21: same -> similar.

P10L8: give the vertical sampling intervals in distance, not in time (permittivity is assumed to be uniform!)

P11L5: Revise. Reflector is an interface, with zero thickness. Do the authors refer the thickness of the layer bounded by two reflectors? Widen –> thicken??

P11L7-8: I don't follow the logic. The SMB varies so density varies as well near the surface. But the density variations get smaller as it becomes deeper so spatial variability of the depth-integrated feature may not be so big (but I don't know whether it can be very small or not).

P11L8-12: At such great depths, not only SMB but also ice flow affect the IRH's shape.

P11L15-: Please reorganize. Bed topography is completely out of the context, and it is indeed confusing. If necessary please change the section's name.

P11L31-33: This ice is called the "echo free zone." If your interpretation is correct, this ice is not useful to reconstruct paleoclimate. It seems not consistent with the great success of the EDC ice core...

P12L5-7: I don't follow where your confidence comes from. Table 1: It is useful if the table includes some features (center frequency, bandwidth, resolution) of the modeled radar data. Also, it is helpful if the table includes references of individual datasets, flight heights, lateral sampling intervals, etc.

Figure 1: Is the CReSIS line continue behind the inset?

Figure 2: unit of the lower panel is probably micro S/m. See my comments about the exponential trend.

Figure 3: please include the firn correction in the main text, and present all relevant information together. Rescale the INGV dataset so that horizontal structure is more visible, I think. What is "extended focused"?
* * *

---

## Author Comment (AC1) · 21 Sep 2016

**Author comments:** "Radio-echo sounding measurements and ice-core synchronization at Dome C, Antarctica"

A. Winter, et al.

5  Review by Tom Jordan, University of Bristol

Overall the scientific analysis is of a high quality and the manuscript is well written. I have, however, made a few suggestions where more detail and precision in the presentation is required; particularly in the methods section. The manuscript is well structured and referenced, with informative figures and tables. Regarding the novelty of the study, I think it needs to be made more explicit as to what differentiates the manuscript from Cavitte et al. 2016 (which also considers IRH detection at Dome C for different radar systems). I appreciate that there are differences, (e.g. the use of the synthetic trace in this study), but this may not be obvious to the general reader. The electromagnetic modelling framework for the synthetic trace appears well established, and physically rigorous. However, given the overall emphasis on comparing how radar system characteristics influence the sounding data, one major area which requires explicit investigation is frequency dependence (see specific comments, Sect. 2.2, Sect. 5.1). Whilst probably not directly impactful for Oldest Ice/synchronization, frequency dependence could be important for combining IRH data sets where reflection amplitudes are important, and thus is required to complete the overall compatibility aspect of this study. The manuscript is clearly of general interest to the readership of The Cryosphere. RES of IRHs can provide useful glaciological information that goes substantially beyond ice core chronologies; for example temperature (from attenuation) and ice dynamics (from IRH derived metrics such as the continuity index). Such a comprehensive compatibility study between different radar systems is therefore of suitably high impact, and will no doubt be a central reference point for future studies which combine RES data sets. Whilst I appreciate that the glaciological application here is Oldest Ice, I have made a few specific comments regarding how the authors could broaden the scope of their introduction and discussion.

*We thank Tom Jordan for his very detailed and constructive comments and readily follow his suggestions to broaden the introduction and discussion of our study. As suggested, we included further glaciological applications, emphasized the novelty of our study with respect to Cavitte et al.'s study, and included the frequency dependence. A comprehensive investigation of reflection amplitudes with respect to frequency dependence was not feasible, the reasons for which we discuss in our answers on the specific comments. We appreciate that some parts of our method were not described with sufficient precision and we have rewritten these sections substantially. Below we address the specific and minor comments one by one with the repeated referee's comment, followed by our answer highlighted in blue italic font.*

**Specific comments**

A clear case needs to be made in the introduction what differentiates the study (both in terms of the questions addressed and the methods that are used) from Cavitte et al. 2016.

*We added a section to the introduction that emphasizes the novelty of our study, especially with respect to the Cavitte et al. (2016) study. We see the major differences in: Cavitte et al.'s major point is to trace the radar stratigraphy to extend the ice core chronology through the ice sheet and possibly connect two deep drill sites for timescale synchronization. To achieve this they combine data of the HiCARS system (new surveys focused on Dome C region) with those of the older UT/TUD survey focused on the Vostok region. Additionally they use one line of the CReSIS MCoRDS system as an alternate route for the single 200 km long TUD connecting line. This separate radar system also serves as additional test of the accuracy of dating of HiCARS reflections at Dome C. The goal of our study is not to trace layers/extend the age scale as far as possible, but to compare the measurements, spatially confined to a small area, of the RES systems that measured a good part of Antarctic RES surveys for combined usage. We compare measurements of FIVE different systems, all within about 2 km. For assessment of the physical properties recorded by the RES systems and their relation to the radar data, and for depth conversion of IRHs we model synthetic radar traces based on the EDC ice core record.*

As mentioned in the general comments, I think the emphasis in the introduction on Oldest Ice is too narrow. This is an excellent opportunity to communicate to the wider glaciological community what rich information is present within RES data (in particular derivable from IRHs), and therefore could be exploited on a broad scale if different data sets can be combined. Two relevant examples are: depth-averaged temperature (from the attenuation rate inferred from internal reflections), Matsuoka et al. 2010, Macgregor et al. 2015b, ice dynamics (from the IRH continuity index) Karlsson et al. 2012.

*We included further glaciological applications of RES data, like inference of ice dynamics and attenuation rates/temperature, in the second half of the introduction. As we made substantial changes to this section, they can be seen easiest in the revised manuscript, where changes are highlighted in color. However, we would like to keep our focus on Oldest Ice and the internal reflection horizons to keep the study constrained. For the same reason we put less emphasis on the relative and absolute amplitudes (see also answer to comment 5.1.).*

**2.1**

It would be helpful here to provide more information regarding the DEP method and the gamma absorption method. In particular; how some of the underlying assumptions/restrictions of these methods could impact the rest of the investigation. For example, in the case of the DEP method, I think that it is important to note explicitly that the reference frequency (100 kHz) is significantly less than the radar systems ( 100 MHz), which may be important regarding frequency dependence of dielectric conductivity/attenuation (see recent discussion in Macgregor et al. 2015b)

*We propose to introduce the frequency dependence of RES reflections and conductivity by adding the following to in Sect. 1:*

*"The IRHs from conductivity changes, in contrast, can be found throughout the ice sheet. The reflection coefficients of those IRHs are related to changes in the imaginary part of the complex dielectric permittivity and are thus proportional to conductivity changes and inversely proportional to frequency. A change of crystal orientation fabric (COF) is the second reason for reflections in the deeper parts of the ice column, predominating in zones of high shear. In RES measurements the conductivity-based IRHs can be distinguished by the frequency dependence of their reflectivity from IRHs caused by COF and density, which have frequency independent reflection coefficients related to changes in the real part of the complex permittivity (Fujita et al., 1999, 2000). Conductivity itself was assumed frequency independent in the range of RES frequencies, but more recent work implies that its frequency dependence cannot be neglected for e.g., attenuation studies (MacGregor et al., 2015b)."*

*and extend the discussion on synthetic vs. RES data (Sect. 6.2.1) by the frequency dependence of conductivity:*

*"Furthermore, conductivity itself is frequency dependent (MacGregor et al., 2015b). This implies that also the different measuring frequency of the DEP compared to the source wavelet (100 kHz vs. 150 MHz) effects the reflection amplitudes of the synthetic trace. However, the uncalibrated conductivity profile of the Dome C core and the simple model itself do not allow to quantitatively analyze the reflection amplitudes, but only the signature of reflection patterns in our study."*

*However, we do not really see a strong point for taking the different frequencies into account, or the temperature for that matter. The bench used at Dome C and the calibration of the conductivity was not as good as for example at EDML, and the real part of the complex dielectric permittivity was not measured. So all that can be used is the conductivity changes. We cannot get the correct absolute conductivity at Dome C and even if we were to take the frequency into account, and the temperature, yet it still is only an uncalibrated relative conductivity. For that reason it is only the changes in conductivity that have some meaning. We do now emphasize in our methods section that we are only using the conductivity changes.*
*The gamma absorption method is described at length in Eisen et al. (2006) and the papers cited therein. We have now included this reference in the manuscript. However, we do not see any impact of underlying restrictions of this method in connection with radar measurements. More so, as we use the density only for calculating permittivities in the upper 100 m.*

**2.2**

Given the range of radar system frequencies that are considered in this study (60-195 MHz) I think it is necessary to investigate the frequency dependence of the synthetic trace. Note; that this is in the context of thin-film interference (and how frequency/wavelength dependence affects the peaks/relative amplitudes of the synthetic trace) rather than the intrinsic frequency dependence of the dielectric conductivity as mentioned above. Specifically, it would be useful to repeat the synthetic trace analysis, for source wavelets at the CReSIS (195 MHz) and UTIG (60 MHz) centre frequencies. If pronounced sensitivity is demonstrated, then these repeat traces could be used to improve the comparison between the synthetic A-scope trace and the radar A-scope traces in Section 5 and the related discussion in Section 6. I hope, given the excellent EM simulation framework available to the authors, this is request is fairly straightforward to do, and would add a significant value to their investigation.

*The issue with CReSIS and UTIG systems is that they are focusing radars and use extended chirps as source signals, not short pulses. The systems use different processing sequences to get to the A-scope presentation. Implementing that in the forward model is not straight forward and time consuming. We have run the model with short pulse wavelets of different carrier frequencies. This does not improve the horizon matching with any of the RES traces considerably, as, for several reasons, the model is not able to correctly reproduce the reflection amplitudes. The synthetic trace solely mirrors the signature of the conductivity IRHs, i.e. the pattern of reflections. Since it would give only small merit, we decided not to add even more (synthetic) traces to the figures. They already contain a lot of information and would become too confusing. For these reasons we chose one source wavelet that can be compared best with most of the traces at the same time, with respect to vertical resolution, and still be matched relatively easy with measured conductivity peaks to keep the depth determination simple. However, we do now include the frequency dependence and the different measuring frequencies, as well as the different source types in the discussion (Sect. 6.1.2 and 6.2.1).*

On a related note, I assume that the EM simulations assume a monochromatic source? Since the chirped radar systems have both finite and differing bandwidths, this is clearly an important simplification. Finally, is it also correct that the EM simulation method is physically a closer representation of the pulse/burst systems rather than the chirped radar systems? Again this needs to be discussed, along with potential caveats for cross-comparison.

*Yes, we used a monochromatic source with decreasing amplitudes towards the margins of the burst length, (which is, however, shorter as for the radar systems) which is a closer representation of the burst systems. It is a simplification, and there are other simplifications as well. For example the model is 1D (to be able to simulate several km of ice at cm resolution) compared to the 3D ice sheet. For these reasons we do not expect the synthetic trace to look exactly like one of the radar data. Nevertheless, the simulation shows distinct reflections and patterns of reflections at the same TWTs as the radar traces and thus is very useful to connect the radar data with the conductivity record.*

*As mentioned in the previous comment, the forward model does indeed mimic a pulsed radar and not a chirp. We extended the discussion of these limitations.*

**2.3**

Given the overall emphasis on cross-comparison between different radar systems, it is desirable that the authors provide evidence for how robust the method used to determine the permittivity of ice is for other radar systems (only AWI is discussed here). The variance in the obtained values could then be discussed in relation to other components of their investigation.

*It was not our aim to get the exact permittivity value or its uncertainties with our method, as we actually do not need the exact value. Our method of depth conversion with the sensitivity studies with the conductivity profile is independent of the permittivity. We just need some good "first guess" for the permittivity so the reflections in the synthetic trace occur at more or less the same TWT as in the measured ones to match the peaks. In fact, as neither system was exactly at the Dome C drill position nor has the core reached bedrock, there is no possibility to get the true permittivity right. However, the accuracy of the permittivity obtained with our method is high enough to warrant the comparison and conclusions. We chose the AWI data because they have comparably high vertical resolution and the burst system is better represented by the modeled data. Additionally, most RES traces are even further off the drill as the AWI profile. That increases the risk of mistakenly interpreting the spacial variations of reflector depths as uncertainties/differences in permittivity. Furthermore, this method of manually comparing various traces of 40000 ns is quite time-consuming. For that reason we would like to keep it as is, as the accuracy of the permittivity we use*

*is sufficient for our purpose. However, we appreciate from several comments that we did not make our aim of a first estimate of the permittivity clear enough. For that reason we revised the whole section 2.3, as shown in the revised manuscript with changes highlighted in red and blue color.*

**5.1**

5    If possible I would like to see a discussion (and potentially an explanation) for the variation in the relative amplitude of the IRH peaks for the different radar systems. In particular, the relative amplitude of the peaks for the UTIG trace appears to be lower than the other systems of comparable vertical resolution (AWI,CReSIS), and this may potentially relate to frequency dependence. One reason why reflection amplitudes are important is that they are used to determine depth-averaged attenuation rates (and thus information about depth-averaged temperature). Subsequently, even if only preliminary conclusions be made
10   regarding differences in amplitude, this will be useful for future combination studies.

*We appreciate that the differences in amplitude are a very interesting point, especially with respect to the attenuation and temperature in the ice. However, as our focus is on age structure of the ice sheet, we are concentrating on the depths of the IRHs, but not on the amplitudes or decline of amplitudes of the reflections (see answer to comment 1). We do not think we have the means to draw any quantitative conclusions regarding the amplitudes. Most of the data that were provided to us*
15   *for this study were already processed in some way, including deconvolution, pulse shaping and gain adjustment. For some systems we even use differently processed data for the different depth regions (always the best visibility of IRHs in mind). That already has significant influence on the relative amplitudes. We cannot compare the relative amplitudes of differently processed data and separate the effects of measuring frequency and processing. To collect all the raw data and calculate the corrected relative decibel power for all the different radar systems would, in our opinion, be beyond the scope of this study. However, we*
20   *added to our discussion (Sect. 6.1.2) and conclusion, that the differences in relative amplitudes, possibly based on frequency dependence, should be closer examined when using RES data to investigate attenuation rates and the reasons for which we did not examine them.*

Since it is mentioned in the conclusions that AWI, CReSIS and UTIG are likely suitable for combined analysis, it would be useful to see some direct cross-over analysis of the traces. I appreciate from Fig 1 that this may only be possible for AWI and
25   UTIG, but this would act to strengthen the overall conclusions regarding combining data.

*Good point. We included this figure (Fig. 1) as a direct comparison of AWI and UTIG radargrams at the cross-over point of the profiles. We think it is a great example that the data are indeed well combinable as all strong horizons can be traced smoothly across the intersection. But this figure also shows, that the strongest reflections in one profile are not necessarily the strongest ones in the other profile (e.g., H4 and horizon just below). This could be caused by the different source types, frequencies,*
30   *and/or processing of the data. As you already pointed out, this cross-comparison is not possible for the other profiles, due to the lack of more cross-over points in the data available to us.*

**6.1.1**

I think it would be helpful here (or potentially in the caption of table 1) to provide relevant equations regarding the relationship between bandwidth and vertical resolution for the chirped systems (e.g. as supplied in the CReSIS reference).
35   *Thank you for this suggestion. We changed the caption of Table 1 and included: "the vertical resolution, determined as $\frac{kc_0}{2B\sqrt{3.17}}$ for the chirp systems. For the window widening factor k we use 1.53, as given in CReSIS (2016)." and changed the values for the vertical resolution, now including the windowing factor. Furthermore, we extended the table by references for the data and the characteristics of the synthetic data.*

**6.2.1**

40   Is it possible to provide a rough estimate of the radar footprint diameter in the Dome C region? My guess is that as we are dealing with comparatively thick ice this would be toward the upper end (or possibly exceed) the range that is stated.

*You are right, there is comparably thick ice in this region and the systems' footprints at the ice-bed interface exceed the given range of 10–100m scale. E.g., the one for the CReSIS system is about 110 m. However, in this context we are not talking about*

[Figure]

**Figure 1.** Intersection of AWI and UTIG profiles for direct visual comparison. Two examples of identified IRHs are indicated by red arrows. Note that the air-ice reflection is shifted to 5 μs. All strong horizons can be traced smoothly across the intersection.

*the bed, but the IRHs in the ice column. Nevertheless, since we neglect the upper few hundred meters, the 10 m scale is quite understated and we changed it to 100 m scale.*

As with the introduction, I think a clear case needs to be made what distinguishes the conclusions of this study from Cavitte et al. 2016.

I think it also needs to be discussed explicitly in the conclusion how variable vertical resolution (particularly how multiple peaks transition to single peaks as function of vertical resolution), pose challenges for combining data sets. This discussion will hopefully also address other glaciological information derivable from IRHs.

*We checked the whole section in consideration of your suggestions. We included the possible challenges due to variable vertical resolution and the investigation of attenuation rates and temperature as additional glaciological information. Furthermore, we put more emphasis on the comparison of RES data and RES vs. synthetic data, which, in our opinion, distinguishes this study from Cavitte et al.'s.*

**Minor comments**

There are quite a few examples where there is no white space preceding the SI units. (e.g. 0.2m). These should be corrected.
Be consistent with the hyphenation of ice-core/ice core.
There are many instances where use of 'vertical resolution' would be less ambiguous than 'resolution'.

5 *We added white space in front of all units, we now use "ice core" without hyphenation throughout the manuscript, and have added "vertical" in front of resolution, where it was ambiguous.*

Line 17: It would be helpful to add a reference here.
*Augustin et al. (2004) was added as a reference.*

10 For brevity, the final paragraph of the introduction could be dropped.
*The paragraph was removed.*

**2.1**

Line 1: ice: → ice (: should only be used for equation arrays/lists)
*We could not find this phrase in this section and assume the comment was unintentionally copied from the 2.2 section.*

15 Line 9: Reword 'we shortly discuss the input parameter permittivity of ice.
*The sentence was reworded to "we describe how we derive the value for the permittivity of ice that we use in all further proceedings."*

Line 11-13: If possible, please reference the core data (temperature, accumulation, etc.)
*The EPICA Dome C 2001-02, science and drilling teams (2002) was added as a reference, additionally to Augustin et al.*
20 *(2004).*

Lines 18,22: Give units for sigma and rho when they are introduced.
*Units were included*

Lines 24,27: Unit spacing for Xmm
*Done*

25 Line 24: measuring: → measurement
*Changed*

Line 25: comma before rho_ice
*Done*

**2.2**

30 Title: Consider changing to 'Electromagnetic modeling of radar traces'
*Title was changed as suggested*

Line 1: ice: → ice (: should only be used for equation arrays/lists)
*Changed*

Line 4/equation 2: The equation is correct, but the symbols (epsilon,epsilon',epsilon",sigma, omega ) must be introduced
35 correctly in text; see, for example, Eisen, et al. 2004, equation (1). Additionally, epsilon_0 is best described as the 'vacuum permittivity' (the use of ordinary is confusing since 'ordinary permittivity' is used in the context of anisotropic media).
*We changed "ordinary permittivity" to "vacuum permittivity" and introduced the symbols.*

Line 14: 'incorrect ordinary permittivities (the real part of the complex relative permittivity)' → 'incorrect real permittivities'.
*Changed*

40 Line 24: Measuring → measurement
*We could not find this and assume the comment was copied from 2.1*

Line 17: '1D-FD' → '1D-FD (One-Dimensional Finite Difference).'
*Added*

Line 19: Please be more specific about the boundary condition(s). Is it the lower or the side boundaries?

*As the model is 1D the absorbing boundary is also implemented in the only space dimension, which is the depth direction. For clarity we added "in the depth direction".*

Line 20: Please reference the Courant Criterion, and state what it tests for (convergence of the numerical solution).

5   *Courant et al. (1928) was added as a reference, and "that ensures the stability of the numerical calculations" was added*

Line 21: See my specific comment about the frequency dependence/sensitivity of the synthetic trace. It should be clearly started here that there are differing frequencies and bandwidths for the radar systems.

*We changed the section to "... we use a source wavelet of two and a half 150 MHz cycles. It should be noted here, that, for simplicity this wavelet is based on the burst and pulse radar systems rather than the chirp systems, which require additional*

10   *post processing like pulse compression (Sect. 3). However, this synthetic pulse is much shorter and the wavelet is not identical to any of the RES system ones. We chose it as trade-off between being long enough to..."*

Line 24: Please provide a reference for Hilbert magnitude transform.

*Hilbert (1906) was added as a reference*

15 **2.3**

Title: Consider changing to 'Determination of the relative permittivity of ice'

*Since we rewrote the whole section, we decided to change the title to: "Assessing the permittivity of ice"*

**3.2**

Line 17,22: Unit spacing for X microseconds.

20   *Done*

**3.4**

Line 27: Unit spacing for km.

*Done*

**3.5**

25   Line 27: Unit spacing for MHz.

*Done*

For completeness it would be helpful to list the distances of the profiles from Dome C for all radar systems (this is only provided for AWI and CReSIS).

*The distance to drill site is given for all systems in the systems' sections. It just is not always in the last sentence of the section.*

30 **4**

Line 15: No new paragraph?

*Changed*

**5.1**

Line 12: second → upper right

35   *Changed to upper middle*

Line 18: resolution → vertical resolution

*Changed*

**5.2**

Line 6: y → vertical
*Changed*
Line 8: about → an approximately
*Changed*
Line 17: is starting → starts?
*Changed*
Line 32: missing → missing from the synthetic trace
*Added*

**5.3**

Line 2: Is there a suitable reference for the 'sensitivity approach' used here?
*Eisen et al. (2006) was added as a reference*
Line 15: The advantage → The advantage of the sensitivity approach. . . ?
*Added*

**6.1.1**

Title: Consider changing to 'Vertical and horizontal resolution'
*Title was changed*
Line 14: UTIG systems the→ UTIG systems to be the?
*Changed*
Line 15: Obviously in → Due to their lower vertical resolution
*Sentence was changed*
Line 25: continue → continue with
*Changed*

**6.1.2**

Line 3: remove 'and others'
*Removed*
Line 8: accumulated → accumulates?
*Changed*
Line 10: it is → the slope is
*Changed*
Line 13: Urbini et al. needs a date.
*We have removed this sentence in the revised manuscript.*
Line 14/15: Reword sentence starting: 'That would...
*We removed the sentence*
Line 22: they → this
*Changed*

**6.1.3**

Line 8: comparing → comparative
*Changed*

**6.2**

Lines 12-14: 6.2.2. and 6.2.1 are introduced in the wrong order.
*Order was changed*

**6.2.1**

5   Lines 19,20,28: Unit spacing for Xm
*Added*
Line 23: examine → who examine
*(Line 22?) This would lead to an incorrect sentence?*
Line 28: find → found
10  *Changed*

**6.2.2**

Line 14: missing full stop after trace
*Added*
Line 15: remove 'used'?
15  *Removed*
Line 27: example: The → example, the
*Changed*

Line 9/10: It probably best to relate the well resolved IHRs for these systems directly to their better vertical resolution than the
20  other systems (rather than implicitly through their bandwidth).
*We changed the sentence to "If interested in well resolved IRHs at intermediate depths, the AWI, CReSIS, and UTIG systems, are the most suitable, due to their comparably high vertical resolution."*
Line 13/14: Reword: The best quality in imaging the basal layer have the CReSIS, UTIG and BAS data, the latter, however, with ...
25  *Changed to: "The CReSIS, UTIG and BAS systems have the largest penetration depth and are able to image some structures in the basal region. Over the comparably low vertical resolution of the BAS data, we attest the CReSIS and UTIG systems the best suitability in our comparison..."*
Line 19: Remove 'profound' (it is best practice to avoid superlatives)
*Removed*
30  For the conclusions it is best practice to use past tense rather than present tense, and I would recommend carefully checking this section.
*We changed the tense of the section to past tense and checked it again. Changes can best be seen in the revised manuscript with color-highlighted changes.*

**Figures**

35  **Table 1**

Relabel 'resolution' as 'vertical resolution'. For completeness it would be desirable to provide an indication of how windowing/processing affects the vertical resolution (e.g. for the CReSIS system I believe that the post-windowed vertical resolution is 4.3 m).
*"Vertical" was added. Furthermore we changed the caption and included the equation for the vertical resolution, including*
40  *the windowing/processing and changed the values accordingly.*

**Fig 2.**

Given that the reflections are ultimately caused by discontinuities in conductivity (rather than the peaks themselves), I think it would be useful to provide a plot of the vertical gradient of conductivity underneath the conductivity plot. I wouldn't be surprised if this gradient plot has a more 'immediate' correspondence with the synthetic trace reflections, and could be used to improve the analysis in Section 5 and 6.

*We replaced the plot of the conductivity with the vertical gradient of the conductivity (for clarity we plotted the envelope). Like you anticipated, the synthetic trace is closer resembled by the gradient than by the conductivity profile*

Following the terminology in Section 4, it should be added explicitly to the caption that Fig. 2 is an A-scope plot

*Caption was changed to "A-scopes for traces of the five radar systems and synthetic trace. ..."*

**Fig 3.**

The font size for the axes labels/numbers should be increased

*The font size was increased.*

of → for

*Changed*

x-axis → horizontal axis

*Changed*

closer →closely?

*Changed*

**References**

Augustin, L., Barbante, C., Barnes, P. R., Barnola, J. M., Bigler, M., Castellano, E., Cattani, O., Chappellaz, J., Dahl-Jensen, D., Delmonte, B., et al.: Eight glacial cycles from an Antarctic ice core, Nature, 429, 623–628, doi:10.1038/nature02599, 2004.

5    Cavitte, M. G., Blankenship, D. D., Young, D. A., Schroeder, D. M., Parrenin, F., Lemeur, E., Macgregor, J. A., and Siegert, M. J.: Deep radiostratigraphy of the East Antarctic plateau: connecting the Dome C and Vostok ice core sites, Journal of Glaciology, pp. 1–12, doi:10.1017/jog.2016.11, 2016.

Courant, R., Friedrichs, K., and Lewy, H.: Über die partiellen Differenzengleichungen der mathematischen Physik, Mathematische Annalen, 100, 32–74, doi:10.1007/BF01448839, 1928.

CReSIS: Radar depth sounder readme, ftp://data.cresis.ku.edu/data/rds/rds_readme.pdf, 2016.15.03, 2016.

10    Eisen, O., Wilhelms, F., Steinhage, D., and Schwander, J.: Improved method to determine radio-echo sounding reflector depths from ice-core profiles of permittivity and conductivity, Journal of Glaciology, 52, 299–310, doi:10.3189/172756506781828674, 2006.

Fujita, S., Maeno, H., Uratsuka, S., Furukawa, T., Mae, S., Fujii, Y., and Watanabe, O.: Nature of radio echo layering in the Antarctic ice sheet detected by a two-frequency experiment, Journal of Geophysical Research, 104, 13 013–13 024, 1999.

Fujita, S., Matsuoka, T., Ishida, T., Matsuoka, K., and Mae, S.: The Physics of Ice Core Records. chap. A summary of the complex dielectric

15    permittivity of ice in the megahertz range and its application for radar sounding of polar ice sheets, 2000.

Hilbert, D.: Grundzüge einer allgemeinen Theorie der linearen Integralgleichungen. Vierte Mitteilung, Nachrichten von der Gesellschaft der Wissenschaften zu Göttingen, Mathematisch-Physikalische Klasse, 1906, 157–228, 1906.

MacGregor, J. A., Li, J., Paden, J. D., Catania, G. A., Clow, G. D., Fahnestock, M. A., Prasad Gogineni, S., Grimm, R. E., Morlighem, M., Nandi, S., et al.: Radar attenuation and temperature within the Greenland Ice Sheet, Journal of Geophysical Research: Earth Surface, 120,

20    983–1008, doi:10.1002/2014JF003418, 2015b.

The EPICA Dome C 2001-02, science and drilling teams: Extending the ice core record beyond half a million years, EOS Transactions, 83, 509, 2002.

---

## Author Comment (AC2) · 21 Sep 2016

**Author comments:** "Radio-echo sounding measurements and ice-core synchronization at Dome C, Antarctica"

A. Winter, et al.

Review by Anonymous Referee #2

This manuscript presents a first comparison of radar data collected with five different systems in the vicinity of Dome C. All of these airborne radar systems (AWI, UTIG, CReSIS and BAS) have generated the majority of the radar data in Antarctica and Greenland so that I see an overall merit to compare these datasets, not only for the oldest ice site survey, but for the ice-sheet research in general. However, the comparison presented in this manuscript is not at all rigorous. It is more or less just a visual inspection to develop fuzzy impressions (that many people already have, I believe), rather than a careful, scientific comparison to rigorously see what can be said and what should not be said by synthesizing different radar datasets together. The goal of the analysis is to compare the RES and synthetic radar data in terms of identifying distinct reflectors that can be found in all datasets and that can be confidently be matched in between the different datasets (P6L10-12). The analysis presented in this paper is inadequate to make this point. As I point out below, I see many not-well-justified procedures in the data/method section. Also, relevant information are shown in many places in the paper, so it is very hard to develop a confident understanding on the analysis presented here.

*We thank the reviewer for their very detailed comments. We agree with the majority of the suggestions. In order to address the concern about the not-well-justified and in-many-places-distributed method description we have substantially revised the sections 2.3 and 5.3. Furthermore we have relativized our statements of what can be found in all, or is comparable in some of the data sets. Below we address all of the individual comments, with our answers italicized in blue, to provide feedback on how we included the suggestions in the revised manuscript. For the few cases where we don't follow the suggestions, we discuss our reasons.*

**Individual comments:**

Titel: With this title, readers cannot find that there is the first comparison of the radar data collected with five different systems. How about "Comparisons of radar data collected by different systems to synthesized radar data using the Dome C ice core, Antarctica"?
*We see your point that the title neglects the comparison part. However, the title you suggest does not clearly include the comparison of the RES data among each other. So we decided to change it to "Comparison of measurements from different radio-echo sounding systems and synchronization with the ice core at Dome C, Antarctica"*

P1L3: bedrock -> bed, bed can be sediment, and not always rock.
*Changed*

P1L4: quality -> capacity?
*Changed as suggested*

P1L10-13: please improve the manuscript, otherwise the statements here are not well supported.
*We revised the sections 2.3 and 5.3 in our manuscript (this is best to be seen in the revised manuscript, where the changes are highlighted in blue and red color) to clarify our motivation and approach for the determination of the permittivity and the depths of IRHs. Furthermore we changed the statement to: "Then we conduct a sensitivity study for which we remove certain peaks from the input conductivity profile. As a result the respective reflections disappear from the modeled radar trace. In this way, we establish a depth conversion of the measured travel-times of the IRHs. Furthermore, we used these sensitivity studies to investigate the cause of observed reflections."*

P1L13: Add "EDC's" before AICC2012. Is it necessary to be so specific on the timescale in the abstract?
*"EDC's" was added and the specific time scale removed.*

P1L17: perfect -> valuable. Nothing is perfect.
5  *Changed*

P1L18:"air bubbles and hydrates"
*"hydrates" was added*

10  P2L3 revise "this new, older core". The oldest core is not drilled yet.
*We made the revision: "this new, older core" -> "this future core". But it is also stated in the two sentences before (P2L1) that this older core is a scientific goal and has not yet been drilled. And, in our opinion, it is made clear by the part "As compared to the oldest continuous ice CURRENTLY on record (retrieved at Dome C...)"(L2) that no older core has been drilled yet.*

15  P2L8: I understand that age structure refers three-dimensional age distribution within the ice sheet. However, it is not necessary to identify the oldest ice. Please clarify what is needed for the oldest ice survey and separately for more general interests of the ice-sheet research
*We revised the section of what is needed for oldest ice survey to: "As many conditions have to be fulfilled at a site for old ice to exist and, equally important, to be retrievable in an analyzable way, extensive pre-site surveys are necessary to fill in gaps*
20  *in the already existing data sets. Of great importance are not only ice thickness and internal structure, but also surface and basal mass balance, ice flow history, as well as temperature profile and geothermal heat flux. Since not all of these parameters are easy to determine in the field, modeling studies will be engaged to constrain upper and lower bounds on parameters which cannot be measured."*

25  P2L9: delete "dielectric properties in front of "density".
*"Dielectric properties" was deleted*

P2L10-17: Fujita and Mae (1994, Ann. Glaciol.) is the first paper to present the frequency dependence of reflectivity in the ice. In the East Antarctic inland, the major reflection cause can be acidity or COF, depending on the radar frequency, ice tem-
30  perature, and acidity/COF contrasts (Fujita et al., 1999 in author's reference list). In general, acidity-based reflection is more dominant at lower frequencies than 50-100 MHz, and COF-based reflection is more dominant at higher frequencies than 100 MHz. Such radar frequency dependence should be briefly mentioned here. And in a later section, the authors should address how 60 MHz data (UTIG) and 150-200 MHz data (other systems) can be compared, even if reflection causes are not necessarily identical.
35  *We agree that we have so far neglected the frequency dependence in our manuscript. As suggested, we now mention it in the introduction by including, how the different reflection causes can be separated. Furthermore we extended our discussion on the differences in the various RES data/between RES and synthetic data (Sect. 6.1.2 and 6.2.1) by the influence of different measuring frequencies on reflection amplitudes.*
*We propose to change this section to: "The IRHs from conductivity changes, in contrast, can be found throughout the ice sheet.*
40  *The reflection coefficients of those IRHs are related to changes in the imaginary part of the complex dielectric permittivity and are thus proportional to conductivity changes and inversely proportional to frequency. A change of crystal orientation fabric (COF) is the second reason for reflections in the deeper parts of the ice column, predominating in zones of high shear. In RES measurements the conductivity-based IRHs can be distinguished by the frequency dependence of their reflectivity from IRHs caused by COF and density, which have frequency independent reflection coefficients related to changes in the real part of*
45  *the complex permittivity (Fujita et al., 1999, 2000). Conductivity itself was assumed frequency independent in the range of RES frequencies, but more recent work implies that its frequency dependence cannot be neglected for e.g., attenuation studies (MacGregor et al., 2015b)."*

P2L16: COF-based reflections do not necessarily constitute isochrones by definition, but Fujita et al. (1999) argued that COF

contrasts can possibly be initiated by acidity contrasts so that regardless of the reflection cause IRH detected at any frequencies can be used as isochrones. This view is supported by a wide range of agreements between modeled isochrones and IRH observed at different radar frequencies. This work also supports this view.

*The sentence stating that COF-based IRHs are not necessarily isochronous was removed by the changes proposed in the answer on the previous comment. However, we do not fully agree with your comment. COF-based IRHs might be formed preferably along acidity contrasts and be isochronous on a large scale. But isochronicity is no necessity for COF-based IRHs to be formed. So we think that our statement was correct, after all.*

P2L19: change to "from any ice core, if the isochrones..."
*Changed as suggested.*

P3L7: "In the sections below we describe the ICE-CORE data used for..."
*"ice core" was added*

P3L11-14: Please add adequate references to characterize EDC core sites. I don't think that Augustin et al. (2004) alone shows the full range of information presented here.
*"The EPICA Dome C 2001-02, science and drilling teams (2002)" was added as a reference.*

P3L19: If I understand correctly, ice temperature is assumed to be -15oC uniformly throughout the core. It is not the case. If the authors just need to have conductivity contrasts to identify acidity-based IRH depths, please say so clearly here to justify the uniform temperature assumption. Anyway, I cannot really understand the motivation of quite complicated (CPU expensive) modeling under such extremely simple assumption.

*Yes, we did not correct for the borehole temperature for exactly the mentioned reason that we just need the conductivity contrasts. Or, as a matter of fact, we cannot do better than only using the conductivity contrasts as the measurements conducted at the Dome C core do not allow for using the absolute values. The bench used at Dome C and the calibration of the conductivity was not as good as for example at EDML, and the real part of the complex dielectric permittivity could not be measured. So all that can be used are the conductivity changes. We cannot get the correct absolute conductivity and even if we were to take the temperature into account, yet it still is only an uncalibrated relative conductivity. The use of a "synthetic radargram" might have raised exaggerated expectations. To clarify: We are not reproducing the correct amplitudes with the model as we do not account for temperature or the correct absolute conductivity and do not use the same source as for the RES data. And also the model is simplified, e.g., one-dimensional. What we can reproduce and what we are interested in is the signature of the (conductivity-caused) IRHs. We will include this aspect in the description of the modeling. To explicitly mention that we just need conductivity contrasts, we changed the sentence in P4L11 to: "However, for the purpose of reproducing the signature of the acidity-caused IRHs as measured by radar, not the absolute value but the changes of conductivity are important." We still use the modeling to account for the interference of reflections of multiple densely spaced conductivity contrasts and for the depth to TWT conversion of the ice core data (further discussed in answer on your comment P4L10-11).*

P3L20: why can the conductivity at the surface be assumed as 4.05 micro S/m? This interpolation is made only for about 7 m so it does not make any major difference. However, I cannot see a reason why the conductivity at the surface has this value. If it is not well justified, why don't you assume it constant over this 7-m-long segment?
*To our knowledge this was the value of the blank measurement of the DEP. We agree to your comment that a constant value would be more appropriate here. But, as you already pointed out, this would not cause any difference in the positions of the reflections in the synthetic trace. It is only a short segment and we are neglecting the upper few hundred meters in our comparison anyway.*

P3L27: add space to 5mm
*Changed.*

P4L2: Is equation(2) necessary?

*Equation(2) describes a variable from equation(1), shows that the dielectric permittivity is a complex value, that the loss factor is related to conductivity and that the imaginary part of the complex dielectric permittivity is inversely proportional to frequency. This all is worth to know for the reader.*

5 P4L8: permittivity is assumed to be 3.17; any reference? Is it reasonable for the temperature range measured at Dome C? Please present an ice temperature profile measured in the borehole.
*We used 3.17 only as a best estimate and not as accurate value for the permittivity. Section 2.3, which is also referenced here, describes how we derived this value. But we do appreciate from several comments that this is hard to understand from our manuscript and thus we revised the whole section 2.3 (see revised manuscript with marked changes). Nevertheless, 3.17 is a*
10 *reasonable value. It is close to the referenced (Rodriguez-Morales et al., 2014; Bohleber et al., 2012) values of 3.15 and 3.16 and Cavitte et al. (2016) also used a permittivity value of 3.17 for their study in the Dome C region.*

P4L10-11: I don't agree. IRH is a result of many individual reflections caused at conductivity contrasts located close to each other. To calculate such interference of many reflected waves, phase is important and phase is dependent on conductivity (its
15 value, not only the contrasts). If author's argument is really the case (i.e. only conductivity contrast is necessary), the authors can simply use the DEP results without any modeling. The bottom line: please clarify what " for the purpose of reproducing reflections" really mean
*As described in the answer to a previous comment (P3L19), the accuracy of the DEP measurement at the Dome C core does not allow for the inference of the absolute conductivity values. Still, using the Maxwell-based model makes a difference compared*
20 *to only using the impulse response to the DEP. The interference of of reflections of densely spaced conductivity contrasts, as also measured by the radar, is reproduced. Eisen et al. (2006) showed that sometimes several conductivity peaks have to be removed from the input conductivity profile in order to completely remove one single peak from the modeled radar trace. This can also be seen in our Fig. 2, e.g., for the H4 reflection. Another advantage of the modeling over using only the DEP results is that the DEP is in the depth domain, whereas the radar measurements are in the TWT domain. Using emice is a good way*
25 *for bringing the conductivity record of the ice core into the TWT domain of the radar data. These are the reasons for using emice and so this is also the answer on your comment P4L21, where you argue for the unnecessity of the modeling. We revised the sentence in the manuscript as follows: "However, for the purpose of reproducing the signature of the conductivity-caused IRHs as measured by radar, not the absolute value but the changes of conductivity are important."*

30 P4L11-13: hard to understand; please revise. Radar data are collected in the two-way travel time domain. Do authors want to say "reflections in the depth domaion" (not the TWT domain)?
*We really mean the TWT domain, as we wrote it. We are talking about the synthetic trace here, which can be seen as our means to convert the ice-core data from the depth domain into the TWT domain of the radar data. Doing so, we only have to convert one data set (the ice core) rather than converting all radar data sets. If we used incorrect permittivities to calculate the*
35 *synthetic trace then the reflections were at the wrong positions in the synthetic trace (in the TWT domain), as compared to the measured radar traces. We suggest to change the sentences to: "Though reflections occur at the wrong TWTs in the synthetic trace when incorrect real permittivities are used in the model, we avoid these errors ..." for a better understanding.*

P4L14: Add space, 0.2 m
40 *Changed.*

P4L14-15: It is a reasonable approach, but explicitly say that in this way only conductivity-based reflections are modeled, and permittivity-based (i.e. COF and density based) reflections are not modeled (Fujita and Mae, 1994, AGlac).
*Yes, that is right. This is already induced by the fact that we are using a constant permittivity below the depth where the den-*
45 *sity reaches the density of ice. For that reason we added the sentences: "Below the depth where the density of ice is reached we use the constant permittivity $\varepsilon' = \varepsilon'_{ice}$. It should be noted at this point that only conductivity-caused reflections and no permittivity-caused (i.e. COF and density based) reflections can be modeled in this way (Fujita and Mae, 1994)." already after line 9. However, the part of the synthetic trace where the permittivity was smoothed is neglected in the comparison of reflections.*

P4L19: model's depth/time increments are interchangeable in this context, I believe. 0.02 nsec and 20 mm are not equivalent if the propagation speed is for e=3.17.

*The depth/time increments are not interchangeable with respect to the propagation speed. It is two different model parameters that are technically independent from each other. But they have to fulfill the condition that $\Delta x / \Delta t$, which is the information propagation speed of the algorithm, has to be greater than the physical wave speed. This condition is exactly the Courant Criterion. Besides do we not use a constant propagation speed for e=3.17 for modeling the synthetic trace. As stated in P4L6-8 and L17, we use the measured densities to calculate the permittivities (Eq. 3) above the firn-ice transition, where we consequently get a higher and depth dependent propagation speed. This is also the reason for which we do not need a firn correction when calculating the synthetic trace or comparing synthetic and measured traces (referring to your comment P4L32). The modeling already accounts for the higher velocities in firn.*

P4L21: What's the exact purpose to use EMICE? Ice temperature is assumed uniform. Permittivity is smoothed out. Input radar waveform to EMICE is inconsistent with any of the radar system. Is it really necessary to use EMICE? Why is the model output smoothed over 150 nsec (equivalent to about 13 m in ice)? If so, please use the model for more realistic conditions.

*The reasons for using emice are explained in our answer on comment P4L10-11. We smoothed the synthetic trace, so that its appearance is better comparable with the measured RES data. However, we changed the smoothing filter to 100 ns. The trace is now a closer representation of the higher resolution RES data. The conditions are realistic enough to reproduce the depths of conductivity-caused IRHs for comparison with the radar data, which is exactly what we need it for.*

P4L27-31: This is a major tuning. It is assumed that AWI data (not exactly at the core site) and synthesized results show consistent englacial reflections and permittivity is tuned. Such assumption should be mentioned more explicitly. What is the corresponding ice temperature to e=3.17? What is the range of permittivity along the core associated with the ice temperature variations? Overall is e=3.17 a reasonable assumption here?

*We understand that we need to better explain this approach and therefore revised the whole section. In our answer on comment P4L8 we already mentioned that 3.17 is indeed is a reasonable value for the permittivity of ice in this region. Our proposed new Section 2.3 is:*

**2.3 Assessing the permittivity of ice**

*To calculate the correct TWTs for reflectors in our synthetic radar trace we have to use the correct permittivities. For too small permittivities the wave speed is too high and a distinct reflection does thus appear too early (Eq. (1)). This time shift increases with the absolute depth of the reflector. As the real permittivity could not directly be measured at Dome C, we are looking for an average value for the permittivity below the firn–ice transition $\varepsilon'_{ice}$ that best reproduces the reflection TWTs compared to measured RES data. Above the firn–ice transition we use measured densities to calculate permittivities, as described in Sect. 2.2. As reference RES data we choose the AWI data for their small distance to drill site, high vertical resolution and being a burst system which is closer represented by the source wavelet of the model than the chirp systems (see Sect. 3 on RES data). Note that it is not our aim to get the exact value of $\varepsilon_{ice}$ but rather a good estimate with this method, so we can easily match reflection peaks of all measured RES data with the synthetic trace in a later step. The exact permittivity is not needed throughout our study because we do not use a velocity function to calculate the depths of RES IRHs but a sensitivity study with the synthetic trace (Sect. 5.3).*

*With the trial-and-error method we compare the synthetic traces of model runs with different $\varepsilon'_{ice}$ to the AWI trace, starting with the commonly used value of $\varepsilon'_{ice} = 3.15$ (e.g., Rodriguez-Morales et al., 2014). We compare the TWTs of ten distinct reflections distributed between approximately 2.3 μs and 24.3 μs TWT between synthetic and AWI trace. This synthetic trace shows smaller TWTs than the measured one, with increasing time lags towards greater TWTs. For this reason we repeat the procedure with an $\varepsilon'_{ice}$ increased by 0.01, and so on. The best result is obtained with $\varepsilon'_{ice} = 3.17$ for which we do not get TWT lags that are systematically changing with increasing TWT between synthetic and measured radar traces and for the compared IRHs. Therefore we conclude that 3.17 is a suitable estimate for $\varepsilon'_{ice}$ in our study region and we will use the synthetic trace calculated with this value for our further proceedings. This value is also found reasonable by Bohleber et al. (2012) for slightly anisotropic configurations and is close to the pure isotropic ice value of $\varepsilon'_{ice} = 3.16$ found in their laboratory experiments.*

P4L32: which IRH are compared between AWI's data and synthesized results? All IRH?? Revise "For this value, the identified reflections occur at the same TWT for both traces".

*We compared 10 reflections that are striking or in a striking pattern. It is not necessarily the afterwards identified IRHs, but some do coincide. We appreciate that we have not explained our proceeding sufficiently and revised the whole section 2.3 (see previous comment). The sentence you mentioned was changed to: "The best result is obtained with $\varepsilon'_{ice} = 3.17$ for which we do not get TWT lags that are systematically changing with increasing TWT between synthetic and measured radar traces and for the compared IRHs. Therefore we conclude that 3.17 is a suitable estimate for $\varepsilon'_{ice}$ in our study region and we will use the synthetic trace calculated with this value for our further proceedings."*

Again, permittivity is temperature dependent. And, somewhere further down in the manuscript, firn correction of 10 m is made (Fig. 3 caption). Then I am really puzzled; information necessary to understand the TWT/depth conversion scattered many places in this paper so it is really hard for me to follow author's logic.

*As explained in our answer on comment P4L19, a firn correction is not needed for this step. We used the measured density down to 113 m plus an extrapolation to the density of ice to calculate the permittivities for the model. But we infer from this and several other comments, that some parts of our method seem not to be described well enough and thus hard to understand. From your comments we get that it is mainly the TWT/depth conversion and the determination of (a first estimate of) the permittivity of ice, and our motivation to use these methods. For this reason we revised these sections, trying to clarify what we are doing and for what reasons.*

P5L8: "(CReSIS) at the University of Kansas, ..."
*"at the University of Kansas" was added*

P5L11ff: Please use Table 1 more effectively. The range of information given in the text is also given in the table. References should be added to identify the processing procedure; descriptions here are too vague and brief to have the full understanding of individual datasets. In addition to items currently presented in Table 1, it is useful to show flight height, stacked distance, reference to processing procedure, etc.

*Thank you for this suggestion. We extended Table 1 by the horizontal trace distance, the original references of the datasets and the characteristics of the synthetic trace. The technical details of the RES data can then be inferred from the papers where the data is presented first, as this is not the focus of our study and is already described elsewhere.*

P5L19: What does "unprocessed" exactly mean?
*We changed the sentence to: "The data are chirp compressed and a horizontal smoothing with a 49 sample moving-average filter and 10-fold stacking is applied."*

P5L28: What is "pulse envelope radar"? Is it a pulse-modulated radar that records only the returned power, not phase?
*Yes, that is right. We changed the sentence: "The INGV profile was measured in December 2011 during a test of a 200ns pulse envelope radar system with a carrier frequency of 150 MHz. "*
*to: "The INGV profile was measured in December 2011 during a test of a 200 ns pulse radar system with a carrier frequency of 150 MHz, recording the envelope only."*

P6L6: What is an automatic gain control? I guess that it is a way to compensate geometric spreading and attenuation within the ice, but no details are given. Is this gain adjusted correctly so that the gain is not increased once the received power reaches to the noise level?
*The automatic gain control is a tool of the processing software that balances the amplitudes in a chosen interval based on statistics. Usually the AGC is applied to the whole recorded trace and does not stop where the noise level is reached. This step was done only for the visual improvement of deeper IRHs.*

P6L10-12: Here, the goal of the analysis is clearly articulated: "our aim is to compare the RES and synthetic radar data in terms of identifying distinct reflectors that can be found in all datasets and that can be confidently be matched in between the

different datasets." However, as I pointed out above and will do so below, the analysis presented here is not adequate to meet this goal.

*Since it is formulated as a goal, we think we can leave the statement as it is at this point. But of course you are right and it is an exaggeration that we can fulfill this as stated. So we relativize the "in all datasets" part at a later point, where your comment P7L13-14 refers to.*

P6L24: Physics behind the sentence "The exponential trend is removed from every trace" is that (1) ice temperature is uniform from the surface to depths, (2) chemistry is also inform, and consequently (3) attenuation rate is uniform. This feature may be seen in the model results, but if so it is only because the ice temperature is assumed to be uniform in the model. Again, I am really puzzled; what do the authors want to replicate by the model and for that goal what can be simplified? Information on these points are scattered many places in the manuscript so it is very hard to read.

*As our focus is on age structure of the ice sheet, we are concentrating on the depths of the IRHs, but not on the amplitudes or decline of amplitudes of the reflections. For that reason there is no need for us to investigate the trend in the RES data. We removed the trend since in that form the positions of the peaks can be compared best. In all our proceedings for plotting the traces and radargrams we always had solely the best visibility of IRHs in mind.*

P6L26-27: Please revise. I cannot understand. Figure 2 shows the returned power in arbitrary scale; is it linear scale or dB scale? If the latter is the case, does the panel show log of log??

*Yes, it is a dB scale, as the received power (in dB) is converted to voltage with a linear relationship before digitization. But it is not normalized in any way and unsuited to derive real amplitudes. However, we changed the figure and are not plotting the log of the amplitudes in the right panel anymore, to get a more consistent presentation. But we still scale the traces with different constants to increase the smaller peaks for visibility. We revised this part to: "The peak amplitudes of the reflections decline in a different manner for the different data, depending on the source wavelet of the radar system and the processing. For that reason, we scale the data differently for the different depth sections to make potential reflections in the basal region more visible."*

P7L1: remove approximately. One third is approximate anyway.

*"approximately" was removed*

P7L10-11: repeated/duplicated information. Delete.

*The sentence was removed*

P7L13-14: I cannot agree at all with the authors. Not all of ten IRH are confidently matched correctly. My quick view found no H6/H10 in AWI, no H3/H6/H9/H10 in UTIG, no H2/H3/H6/H9 in CReSIS, no H1-H6/H10 in INGV, and no H3/H6/H9 in BAS. I don't expect that all of these features match pretty well between the all datasets. This level of agreement is something I don't surprise and it is indeed a new finding of this analysis. Please articulate what you found; don't stretch your results. I did not comment the rest of Section 5.1 and most of Section 5.2; please revise it accordingly based on my suggestions above.

*You are right, of course. Not all ten IRHs can be matched in all of the data and we apologize for this misdescription. Additionally to having written "reflections that can be identified in SOME or all of the data", we now included the sentences "It should be noted here that the peaks do not have the same relative amplitudes or widths in the various data. Furthermore, not all IRHs can be found in all the data." to make this clearer. However we are surprised by the list of IRHs you did not find, as some of them are very distinct, e.g. H3/H9 in BAS, H4/H5 in INGV, H3/H6/H9 in CReSIS, H10 in UTIG (Fig. 2) and indeed are confidently matched in our opinion. Of course, the relative amplitudes of the peaks are different in the different data for the discussed reasons of spatially changing IRHs and there are comparably more reflections included into one peak in the lower resolution data.*

P8L25: the term "echo-free zone" is misused here, if the authors follow the original definition made by Fujita et al. (1999, JGR). It does not simply refer the ice from which no echo is received. It's upper surface is associated with a significant, sharp drop of the returned power, indicating the sudden loss of reflection even if the incident radio wave is strong enough (not attenuated so much).

*Yes, thank you for pointing this out. What we really wanted to say is that in this deep section no coherent IRHs are visible anymore. But of course echo-free zone is not suitable here, as the returned power continuously decreases for the AWI and INGV data and thus the lack of IRHs is an issue of the systems' power. We changed the part to: "...leaving about 800 m without IRHs.". Instead, we now introduce the term echo-free zone in section 6.1.3, following the original definition by Fujita et al. (1999) with citing them (see answer on comment P11L31-33).*

P9L2-4: Again I'm really puzzled. What is the sensitivity test? AWI data were already used to have a best estimate permittivity/propagation speed. Why is conductivity mentioned here, though it is irrelevant to depth estimate? I did not comment on the rest of Section 5.3.

*It seems we do not have explained our method sufficiently and we apologize for that. The permittivity estimated before from the AWI data is only a first estimate for plotting the traces and matching the horizons between synthetic and measured traces. After this step, the sensitivity test with the conductivity is the method we use to determine the accurate depth of the identified horizons. So the conductivity profile is not at all irrelevant to depth estimate. We understand that the name "sensitivity" test might be misleading. We do not use it to test for any amplitudes, but to test if certain reflections still occur in the modeled trace, or not, after temporarily removing sections from the input conductivity profile. From this comment and your P4L32 comment we appreciate that some clarification is needed in the method sections of permittivity determination and depth conversion. We revised both sections 2.3 and 5.3. The proposed first part of the revised Sect. 5.3 is:*

*5.3 Depths of the RES reflectors*

*To determine the depths of the IRHs, identified in Fig. 2 and Fig. 3, we conduct a sensitivity study of the synthetic trace as described in Eisen et al. (2006). By sensitivity study we mean that we remove certain peaks from the measured conductivity profile (bottom panel of Fig. 2) and run the model with the changed input conductivity profile. As a result the respective reflection peaks disappear from the synthetic trace. As the synthetic trace closely resembles the conductivity profile, the conductivity peaks of interest can be identified with relatively small effort. An exception is the very uppermost part (~400 m), where the reflectivity is influenced not only by conductivity but also by density variations.*

P9L29-30: Please make more rigorous discussion. For me, section 5.1 and 5.2 are inadequate to draw this conclusion.

*We changed the statement to "As pointed out in Sect. 5.1 and 5.2, there are some common features notable in all of the RES data and the synthetic trace (e.g.,...)"*

P9L21: same -> similar.

*Was changed in line 31, not found in line 21.*

P10L8: give the vertical sampling intervals in distance, not in time (permittivity is assumed to be uniform!)

*The sampling interval in distance is given right in the next line (L9). We revised this sentence to "This gives one sample every 1.1 m, and 3.8 m, respectively in the ice of the Dome C region". We like to keep it in time as well, as in this form it is more generally valid. It is how the sampling interval is defined for the radar measurements and independent of the material (permittivity) investigated.*

P11L5: Revise. Reflector is an interface, with zero thickness. Do the authors refer the thickness of the layer bounded by two reflectors? Widen –> thicken??

*Yes, thank you. We have misused the word here. In this case we refer to layers of equal dielectric properties, e.g., one ash layer. We revised the sentence to "This can have the effect that some layers may be thicker in one profile than in another, leading to different appearances of the according reflections in the radargrams. It is also possible that some signals are missing completely at one location, ...".*

P11L7-8: I don't follow the logic. The SMB varies so density varies as well near the surface. But the density variations get smaller as it becomes deeper so spatial variability of the depth-integrated feature may not be so big (but I don't know whether

it can be very small or not).

*We apologize for our imprecise formulation. You are certainly right when it comes to random accumulation variations. In that case the amount of snow fallen at different locations spatially evens out over the years. In contrast, in our argumentation we were thinking of accumulation rates that are varying spatially but are more or less stationary over larger time scales of some*
5 *10 ka or even more. That is a completely different issue in terms of the depth-integrated feature, as the depth of IRHs does depend on accumulation rate. For clarification we suggest to change the sentence to: "When stationary, even small spatial accumulation variations cause spatial variations in reflector depths."*

P11L8-12: At such great depths, not only SMB but also ice flow affect the IRH's shape.
10 *Yes, thank you for pointing this factor out. We have neglected the ice flow here because we rated it a minor factor, due to the dome position. But you are right and this was an oversimplification. E.g., Urbini et al. (2008) show that the dome was not always at the same position. And, like for the variations in accumulation rate, even very slow ice flow has a large impact in the vast amount of time we are dealing with. We added the ice flow to the factors influencing the IRHs.*

15 P11L15: Please reorganize. Bed topography is completely out of the context, and it is indeed confusing. If necessary please change the section's name.
*Apart from accumulation rate and ice flow, the age structure of the ice sheet is also influenced by the ice thickness. So we do not think the bed topography is out of context, as we investigate how the differences in reflection depth/TWT in the different radar data could arise from their different profile locations. When the bed topography has elevation variations of several tens*
20 *of meters within one kilometer then this gives ice-thickness changes on the same order, due to the relatively flat surface. And this indeed has substantial influence on the depths of deep IRHs. To better prepare the reader we changed the section's title to: "Spatial variation of IRH depth and strength" and added the sentence: "Since ice thickness is a factor for IRH depths, the slopes of IRHs are also influenced by the bed topography."*

25 P11L31-33: This ice is called the "echo free zone." If your interpretation is correct, this ice is not useful to reconstruct pa-leoclimate. It seems not consistent with the great success of the EDC ice core...
*Our formulation has apparently missed the point, for which we apologize. We removed the misleading sentence with "basal layer" from the manuscript and replaced it with: "This basal region, indicated by a sudden drop of returned power in the radar data is described as echo-free zone (EFZ) by Fujita et al. (1999) and Drews et al. (2009)."*
30 *However, the interpretation as echo-free zone does not mean this ice is useless for reconstructing paleoclimate. As has been clearly demonstrated at the EDML ice core, the onset of the echo-free zone corresponds to that regime, where small-scale (i.e. ice-core diameter) deformation takes place. As Ruth et al. (2007) showed for EDML, the paleo-record is still useful below that depth. (In fact, at EDML the EFZ onset was at 2040 m, useful ice down to 2400 m and the bedrock at 2785 m). Nevertheless, the interpretation might be more complicated than that in the undisturbed regions above - take the NEEM ice-core record as*
35 *an example. The EFZ itself is not sufficient to discard a site - as the ice might still be continuously datable - but the risk of getting a disturbed record is elevated.*

P12L5-7: I don't follow where your confidence comes from. Table 1: It is useful if the table includes some features (cen-ter frequency, bandwidth, resolution) of the modeled radar data. Also, it is helpful if the table includes references of individual
40 datasets, flight heights, lateral sampling intervals, etc.
*That is a good suggestion, thank you. We included the features of the synthetic data and the original references for the RES datasets in Table 1. The technical details of the RES data can be inferred from the papers where the data is described first, as that is not the focus of our study.*

45 Figure 1: Is the CReSIS line continue behind the inset?
*We changed the inset, so the continuing CReSIS line is visible.*

Figure 2: unit of the lower panel is probably micro S/m. See my comments about the exponential trend.
*Thank you for spotting this error. Of course it is $\mu S\ m^{-1}$. The meters must have gone missing at some point. We have changed*

*the figure and are now plotting the gradient of the conductivity, so we changed the unit to $\mu S\ m^{-2}$.*

Figure 3: please include the firn correction in the main text, and present all relevant information together. Rescale the INGV dataset so that horizontal structure is more visible, I think. What is "extended focused"?

5 *The sentence was changed to 2D focused processing, also this part went missing, we apologize for that.*
*In a former version of Figure 3 we had the different panels scaled with profile lengths. But it made the figure somewhat unsettled. As the rescaling did not really improve the horizontal structure of the INGV data we prefer to leave all panels with the same size.*
*We did not include the firn correction in the main text since it is not relevant for any step in our method. It is not needed for*
10 *the estimate of permittivity, because we used the core's density profile to calculate the permittivities in the firn section for the modeling (see also our answer on comment P4L19). And we do not need it for the determination of the horizon's depths, as we do not use any velocity for TWT to depth conversion, but the conductivity profile. We hope this became clear now with the revised sections on permittivity and depth determination of the horizons. This figure's depth axis is the only time we use a velocity and firn correction for depth conversion, as here we prefer a continuous depth scale.*

**References**

Bohleber, P., Wagner, N., and Eisen, O.: Permittivity of ice at radio frequencies: Part II. Artificial and natural polycrystalline ice, Cold Regions Science and Technology, 83, 13–19, doi:10.1016/j.coldregions.2012.05.010, 2012.

Cavitte, M. G., Blankenship, D. D., Young, D. A., Schroeder, D. M., Parrenin, F., Lemeur, E., Macgregor, J. A., and Siegert, M. J.: Deep radiostratigraphy of the East Antarctic plateau: connecting the Dome C and Vostok ice core sites, Journal of Glaciology, pp. 1–12, doi:10.1017/jog.2016.11, 2016.

Drews, R., Eisen, O., Weikusat, I., Kipfstuhl, S., Lambrecht, A., Steinhage, D., Wilhelms, F., and Miller, H.: Layer disturbances and the radio-echo free zone in ice sheets, The Cryosphere, 3, 195–203, doi:10.5194/tc-3-195-2009, 2009.

Eisen, O., Wilhelms, F., Steinhage, D., and Schwander, J.: Improved method to determine radio-echo sounding reflector depths from ice-core profiles of permittivity and conductivity, Journal of Glaciology, 52, 299–310, doi:10.3189/172756506781828674, 2006.

Fujita, S. and Mae, S.: Causes and nature of ice-sheet radio-echo internal reflections estimated from the dielectric properties of ice, Annals of Glaciology, 20, 80–86, 1994.

Fujita, S., Maeno, H., Uratsuka, S., Furukawa, T., Mae, S., Fujii, Y., and Watanabe, O.: Nature of radio echo layering in the Antarctic ice sheet detected by a two-frequency experiment, Journal of Geophysical Research, 104, 13 013–13 024, 1999.

Fujita, S., Matsuoka, T., Ishida, T., Matsuoka, K., and Mae, S.: The Physics of Ice Core Records. chap. A summary of the complex dielectric permittivity of ice in the megahertz range and its application for radar sounding of polar ice sheets, 2000.

MacGregor, J. A., Li, J., Paden, J. D., Catania, G. A., Clow, G. D., Fahnestock, M. A., Prasad Gogineni, S., Grimm, R. E., Morlighem, M., Nandi, S., et al.: Radar attenuation and temperature within the Greenland Ice Sheet, Journal of Geophysical Research: Earth Surface, 120, 983–1008, doi:10.1002/2014JF003418, 2015b.

Rodriguez-Morales, F., Gogineni, S., Leuschen, C. J., Paden, J. D., Li, J., Lewis, C. C., Panzer, B., Gomez-Garcia Alvestegui, D., Patel, A., Byers, K., et al.: Advanced multifrequency radar instrumentation for polar research, Geoscience and Remote Sensing, IEEE Transactions on, 52, 2824–2842, doi:10.1109/TGRS.2013.2266415, 2014.

Ruth, U., Barnola, J.-M., Beer, J., Bigler, M., Blunier, T., Castellano, E., Fischer, H., Fundel, F., Huybrechts, P., Kaufmann, P., et al.: " EDML1": a chronology for the EPICA deep ice core from Dronning Maud Land, Antarctica, over the last 150 000 years, Climate of the Past, 3, 475–484, doi:10.5194/cp-3-475-2007, 2007.

The EPICA Dome C 2001-02, science and drilling teams: Extending the ice core record beyond half a million years, EOS Transactions, 83, 509, 2002.

Urbini, S., Frezzotti, M., Gandolfi, S., Vincent, C., Scarchilli, C., Vittuari, L., and Fily, M.: Historical behaviour of Dome C and Talos Dome (East Antarctica) as investigated by snow accumulation and ice velocity measurements, global and planetary change, 60, 576–588, doi:10.1016/j.gloplacha.2007.08.002, 2008.

---

## Author Response (AR1)

Our point-to-point response to the reviews is given in the two author comments to the referee comments.

The major changes made in the revised manuscript are as follows:

- We broadened our introduction and conclusions by including further glaciological applications

- We extended our introduction, discussion and conclusions by the frequency dependence of conductivity and conductivity-caused reflection amplitudes

- We made substantial changes to the method sections to clarify why we use the methods and to describe our approach more precisely

These and all other changes can be seen in the marked-up manuscript version below:

[revised manuscript text omitted]

---

## Referee Report (RR1)

**Second review of: Comparison of measurements from different radio-echo sounding systems and synchronization with the ice core at Dome C, Antarctica**

Anna Winter et al. 2016, The Cryosphere.

Reviewer: Tom Jordan, University of Bristol, 14th October 2016.

**Summary**

The revised manuscript is much improved, and I now recommend publication to The Cryosphere subject to minor revisions. Below I provide a response to the specific/major comments which I previously raised, along with some additional minor comments.

**Response to previous specific/major comments**

*1. We broadened our introduction and conclusions by including further glaciological applications*

I am pleased that the authors have taken the opportunity to broaden the impact of their study by mentioning other relevant information that can be derived from IRHs. The scientific gains that can be made from data combination are also now clearly described, which is helpful for motivating the rest of the study. The authors have also provided a better explanation regarding what distinguishes their study from Cavitte et al. 2016.

The conclusions are clear that IRH-derived data combination that is dependent upon reflection amplitudes (e.g. attenuation/temperature) requires further investigation, which nicely motivates further studies. I think it may be useful, however, to add a sentence or two about the consequences of the study for data-combination regarding IRH-derived information about ice dynamics.

*2. We extended our introduction, discussion and conclusions by the frequency dependence of conductivity and conductivity caused reflection amplitudes*

In my first set of reviewer comments I mentioned that frequency dependence could be impactful for the study in two different respects: (i) with respect to frequency dependent attenuation/frequency dependent reflection coefficients (i.e. related to intrinsic dielectric properties), (ii) with respect to thin-film interference effects (i.e. related to the optical thickness of the layers in the simulation, and the frequency dependent resonances which occur).

Whilst I think the authors have dealt excellently with point (i) in the revised manuscript, I still think that point (ii) could have be investigated using the simulation framework available (if only drawing a comparison between the relative reflection amplitudes of the synthetic traces for different frequencies, with all other parameters being kept the same). However, given that the authors now give a good discussion why reflection amplitudes both complex to investigate, and are a key issue to address in future data combination work, I think it is ok to leave this out of the final manuscript. As a compromise, it would be helpful to add to the bottom of page 15 (6.2.1) that frequency dependent thin film interference effects could also impact upon reflection amplitudes, and may complicate data combination yet further.

*3. We made substantial changes to the method sections to clarify why we use the methods and to describe our approach more precisely*

The methods (Section 2) is much improved and symbols are now properly defined. The limitations of using the EM simulation method for comparing burst/pulse systems (which the simulation method

best approximates) with the chirped systems is now made clear. The referencing and description of equations/symbols is also greatly improved. Section 2.3 (Assessing the permittivity of ice) has also been substantially revised and the authors have been much clearer about the purpose of sub-investigation.

**Minor comments/typographical errors**

Abstract, L10: Then we → We then

P5, L2: proceeding→ procedure

P5, L7,L14: It may be more useful to provide more recent references for the Courant criterion and Hilbert magnitude transformation. I appreciate to some of the readership these may be common knowledge, but a recent textbook reference could be useful for those wishing to replicate the simulation.

P6, L16: in→ at a

P5, L2: Missing prime on epsilon_ice?

P8, L10: Missing full stop.

P9, L24: I think it should be noted that the peaks in the grad(sigma) plot, represent the greatest discontinuities in the dielectric properties, and hence are associated with the reflection peaks.

P14, L7. Delete `do'

P16, L5.  Reword sentence beginning with `Over'

**Figures/Tables**

Table 1: `Modeled' needs to be realigned

Figure 4: time in micron → time in microseconds (micron has dimensions of length). It may also be helpful to note that the intersection point is ~1 km south of the core in the caption.

---

## Author Response (AR2)

**Author comments on second review by Tom Jordan, University of Bristol, of:**
**Comparison of measurements from different radio-echo sounding systems and synchronization with the ice core at Dome C, Antarctica**
Anna Winter et al. 2016, The Cryosphere.

We thank Tom Jordan for this positive review. We agree that the revised manuscript is much improved, having benefited a lot from his previous constructive comments. Below we describe the additional changes made in response to the second review, with the reviewer's comments in italic font.

**Response to previous specific comments**

*1. The conclusions are clear that IRH-derived data combination that is dependent upon reflection amplitudes (e.g. attenuation/temperature) requires further investigation, which nicely motivates further studies. I think it may be useful, however, to add a sentence or two about the consequences of the study for data-combination regarding IRH-derived information about ice dynamics.*

We added two sentences to the conclusion about how our study contributes to derive information about ice dynamics: "The combining of different RES data and the dating of horizons is requisite for a large-scale mapping of the age structure of the East Antarctic ice sheet, where no reasonable coverage and resolution is obtained with only one data set. The age architecture in turn facilitates the inference of spatial and temporal variations of mass balance and provides boundary conditions or parameters for large scale ice-flow models."

*2. In my first set of reviewer comments I mentioned that frequency dependence could be impactful for the study in two different respects: (i) with respect to frequency dependent attenuation/frequency dependent reflection coefficients (i.e. related to intrinsic dielectric properties), (ii) with respect to thin-film interference effects (i.e. related to the optical thickness of the layers in the simulation, and the frequency dependent resonances which occur).*
*Whilst I think the authors have dealt excellently with point (i) in the revised manuscript, I still think that point (ii) could have be investigated using the simulation framework available (if only drawing a comparison between the relative reflection amplitudes of the synthetic traces for different frequencies, with all other parameters being kept the same). However, given that the authors now give a good discussion why reflection amplitudes both complex to investigate, and are a key issue to address in future data combination work, I think it is ok to leave this out of the final manuscript. As a compromise, it would be helpful to add to the bottom of page 15 (6.2.1) that frequency dependent thin film interference effects could also impact upon reflection amplitudes, and may complicate data combination yet further.*

As suggested we added the following sentences to section 6.2.1: "In addition to these factors, all related to the intrinsic frequency dependence of the dielectric properties, also the frequency dependent thin-film interference is influencing the measured and modeled amplitudes. The signals from interfaces between layers with constant dielectric properties can be strengthened or weakened by constructive or destructive interference, depending on the thickness of the layers and the source frequency. This may complicate any data combination that is dependent upon amplitudes yet further."

**Minor comments/typographical errors**

Thank you for spotting these errors.
We corrected all typographical errors, except we could not find the 'do' on P14, L7.

*P5, L7,L14: It may be more useful to provide more recent references for the Courant criterion and Hilbert magnitude transformation. I appreciate to some of the readership these may be common knowledge, but a recent textbook reference could be useful for those wishing to replicate the simulation.*

Taflove, A. and Hagness, S. C.: Computational electrodynamics: the finite-difference time-domain method, Norwood, 2nd Edition, MA: Artech House, 1995 were added as a reference for the Courant criterion and Hilbert magnitude transformation in E-M finite differences modeling.

*P9, L24: I think it should be noted that the peaks in the grad(sigma) plot, represent the greatest discontinuities in the dielectric properties, and hence are associated with the reflection peaks.*

We added the sentence. "The peaks of the gradient of the conductivity represent the greatest discontinuities in the dielectric properties, and thus are associated with the reflection peaks in the radar data." to section 5.1.

*P16, L5. Reword sentence beginning with 'Over*

The sentence was reworded to: "Nevertheless, due to the BAS data's comparably low vertical resolution, we attest the CReSIS and UTIG systems the best overall suitability in our comparison for Oldest Ice reconnaissance surveys."

*Table 1: 'Modeled needs to be realigned*

Unfortunately we cannot see, how 'Modeled' needs realignment.

*Figure 4: time in micron time in microseconds (micron has dimensions of length). It may also be helpful to note that the intersection point is 1 km south of the core in the caption.*

[revised manuscript text omitted]